# PROVABLE PRIVACY ATTACKS ON TRAINED SHALLOW NEURAL NETWORKS

## ABSTRACT

We study what provable privacy attacks can be shown on trained, 2-layer ReLU neural networks. We explore two types of attacks; data reconstruction attacks, and membership inference attacks. We prove that theoretical results on the implicit bias of 2-layer neural networks can be used to provably reconstruct a set of which at least a constant fraction are training points in a univariate setting, and can also be used to identify with high probability whether a given point was used in the training set in a high dimensional setting. To the best of our knowledge, our work is the first to show provable vulnerabilities in this setting.

## 1 INTRODUCTION

Recently, it was shown that theoretical tools used to study the implicit bias properties of successfully trained neural networks can be leveraged to reconstruct certain portions of the dataset on which the neural network was trained (Haim et al., 2022). The idea behind these attacks is that under some conditions, trained neural networks must satisfy certain properties that are a consequence of the implicit bias of the training algorithm, which can be used to extract information on the training set. This was followed by many other works that applied the same techniques in a broader setting (Buzaglo et al., 2023a;b; Andrew et al., 2023; Ye et al., 2023; Boenisch et al., 2024), raising this vulnerability as a potential practical concern for the widespread use of neural networks. However, despite the fact that these works were motivated by theory, none of them give an explanation for why such a reconstruction is possible, since a given neural network, which satisfies these properties, may have been trained on potentially many different datasets, including some that are significantly different than the actual data the neural network was trained on.

In this paper, we take what is to the best of our knowledge the first step in developing a theoretical understanding of the privacy vulnerabilities induced by the above implicit bias, by showing that such attacks can be provably executed on trained neural networks under various assumptions. This indicates that such attacks are successful since *all* neural networks satisfying these properties must store at least some information on the training data, which can be used by a malicious attacker. More specifically, we use known results on the implicit bias of ReLU neural networks, which establish that such networks tend to converge to a certain margin maximization solution (Lyu and Li, 2020; Ji and Telgarsky, 2020). This characterization of the implicit bias of neural networks allows us to rigorously analyze certain cases in which the neural network memorizes the training data. In particular, this includes examples where an attacker is capable of reconstructing certain portions of the data in a univariate setting, or perform membership inference attacks with high success rates in a high dimensional setting, effectively distinguishing between instances that are in the training set and fresh instances that were generated by the same distribution that was used to generate the training set.

While our attacks are applicable under certain input's dimension, we also conduct experiments that show that these vulnerabilities can be a concern in more generality, even when our assumptions on the dimension of the input are not met. Nevertheless, it is currently not clear what is the extent of the vulnerabilities that we reveal, and to what extent they can be circumvented. We leave the intriguing question of how to provably defend against such exploits to future work, and we hope that our work will pave the way for and motivate additional rigorous study of privacy attacks and defenses in trained neural networks.

The remainder of the paper is structured as follows: After specifying our contributions in more detail below, we turn to discuss related work. In Section 2 we present our notations, some required background, and the main assumptions we make throughout the paper. In Section 3 we study data reconstruction in the univariate setting, and in Section 4 we study membership inference attacks in high dimensions. Lastly, in Section 5, we conduct experiments to empirically support our findings, even in cases where our assumptions do not necessarily hold.

OUR CONTRIBUTIONS

Our main contribution is to provide rigorous guarantees in this setting, since to the best of our knowledge, all previous works are empirical. In more detail, our contributions can be summarized as follows:

- We prove that in the univariate case, under Assumption 2.1, which states that the weights of a trained neural network reach a stationary point of a maximum-margin problem that can be expressed as a function of the training data, an attacker can reconstruct a portion of the training data with a constant probability, which is independent of the training set and the size of the network. We show how to extract that portion of the training data in Algorithm 1.

- We prove that in the high dimensional case, under Assumption 4.1, i.e. that the vectors in the training data are nearly orthogonal w.h.p., an attacker can execute a membership inference attack with high success rates. We show that some commonly used continuous distributions satisfy Assumption 4.1, and we also provide in Subsection 4.1 examples of different attacks that can be performed depending on the information available to the attacker.

- We empirically show that the membership inference attack we analyze in Section 4 may still be executed successfully when we slightly relax Assumption 4.1. This suggests that the vulnerabilities we study in this paper are potentially even more widespread than what our theory establishes.

RELATED WORK

Privacy attacks in neural networks were studied extensively in recent years. Since this paper focuses on two specific types of attacks, we only review here papers that also study these kinds of attacks, or those that closely relate to it.

**Data reconstruction attacks.** Data reconstruction attacks aim to fully recover the training set or parts of it. These include attacks on generative models such as large language models (Carlini et al., 2019; 2021; Nasr et al., 2023), diffusion models (Somepalli et al., 2022; Carlini et al., 2023), and in federated learning settings (Zhu et al., 2019; He et al., 2019; Hitaj et al., 2017; Geiping et al., 2020; Huang et al., 2021; Wen et al., 2022). Perhaps the most relevant works that are concerned with reconstruction attacks are Haim et al. (2022) and Buzaglo et al. (2023a). Using a known result on the implicit bias of trained neural networks, they define and optimize over a loss function, which upon empirical minimization, allows for the recovery of some of the training set. Inspired by these works, we use the same constraints implied by the implicit bias to study this problem, but to rigorously *prove* the existence of privacy vulnerabilities rather than empirically demonstrate them.

**Membership inference attacks.** The second type of attacks we consider in this paper are membership inference attacks (Shokri et al., 2017; Hu et al., 2022a; Olatunji et al., 2021; Shejwalkar et al., 2021), which discern the inclusion or exclusion of a particular data point within the training set. This attack exploits the observation that machine learning models often behave differently on the data that they were trained on versus fresh test examples. One such difference is that trained models tend to output more confident predictions on training examples compared to test examples. This difference can be used to determine if a certain point was in the training set or not. Olatunji et al. (2021) used this confidence-based technique on graph neural networks. Jha et al. (2020); Farokhi and Kaafar (2020) use tools from information theory to upper bound the probability of success of a membership inference attack on neural networks, which is in contrast to our result which exempli-

fies settings with provable lower bounds on the success rates. Attias et al. (2024) also show provable membership inference attacks, but for models whose objective function is a convex function.

**Differential privacy.** A fundamental theoretical framework in the study of privacy is differential privacy (Dwork, 2006; Abadi et al., 2016; Gong et al., 2020; Pannekoek and Spigler, 2021), which is intuitively used to guarantee that sharing some information on a given dataset is done without leaking too much information on specific instances. This framework constitutes a rather strong standard for privacy guarantees, whereas we consider a setting where our assumptions on the implicit bias of neural networks are typically not differentially private. Namely, we study a setting where our base assumption is that there is already some data leakage in terms of differential privacy, and our work explores what is the extent of the information that can be extracted. Thus, our results are not directly comparable to those which study differential privacy.

**Benign overfitting.** Another well-studied phenomenon in the theory of deep learning, which may explain the prevalence of privacy vulnerabilities, is benign overfitting (Bartlett et al., 2020; Cao et al., 2022; Li et al., 2021). This is when a neural network overfits on the training set, essentially achieving perfect training error, but still enjoys very good generalization on previously unseen instances. This suggests that even well-performing neural networks can memorize their training sets, and therefore become more prone to privacy attacks. While this provides a potential theoretical explanation for this phenomenon, as does our work, it does not immediately imply a method for extracting any information on the training set, nor does it prove the existence of such a method.

## 2 BACKGROUND, PRELIMINARIES AND NOTATION

In this section, we introduce the notations and settings used throughout this paper, and discuss relevant background.

We consider a binary classification setting, where each data instance consists of a pair $(\mathbf{x}, y) \in \mathbb{R}^d \times \{-1, 1\}$, and we define the training set as $\{(\mathbf{x}_i, y_i)\}_{i=1}^n$ which consists of $n$ data points. We let $\Phi(\boldsymbol{\theta}; \cdot) : \mathbb{R}^d \to \mathbb{R}$ denote a neural network, where $\boldsymbol{\theta} \in \mathbb{R}^k$ are the parameters of the network represented as a vector. Let $\ell : \mathbb{R} \to \mathbb{R}$ denote the exponential loss function $z \mapsto e^{-z}$ or the logistic loss function $z \mapsto \log(1 + e^{-z})$, and let $L(\boldsymbol{\theta}) := \frac{1}{n} \sum_{i=1}^n \ell(y_i \cdot \Phi(\boldsymbol{\theta}; \mathbf{x}_i))$ be the empirical (training) loss. A network $\Phi(\boldsymbol{\theta}; \mathbf{x})$ is called *homogeneous* if there exists $c > 0$ such that for every $b > 0$, $\boldsymbol{\theta}$ and $\mathbf{x}$, it holds that $\Phi(b \cdot \boldsymbol{\theta}; \mathbf{x}) = b^c \Phi(\boldsymbol{\theta}; \mathbf{x})$. The ReLU activation function is $[x]_+ := \max(0, x)$, and a homogeneous 2-layer ReLU network has the form $\Phi(\boldsymbol{\theta}, \mathbf{x}) = \sum_{j=1}^k v_j \left[\mathbf{w}_j^\top \mathbf{x} + b_j\right]_+$ where $\boldsymbol{\theta}$ encapsulates the parameters $\{\mathbf{w}_j, v_j, b_j\}_{j=1}^k$. We denote the $(d-1)$-dimensional unit sphere in $\mathbb{R}^d$ by $\mathbb{S}^{d-1} := \{\mathbf{x} \in \mathbb{R}^d : \|\mathbf{x}\|_2 = 1\}$. We use standard asymptotic notation (e.g. $O, o, \Omega$, etc.).

The following known result characterizes the implicit bias in homogeneous neural networks, by showing that these networks converge to a critical point of a certain margin-maximization problem.

**Theorem 2.1** (paraphrased version of Lyu and Li (2020), Ji and Telgarsky (2020)). *Let $\Phi(\boldsymbol{\theta}; x)$ be a homogeneous ReLU neural network. Consider minimizing the logistic ($z \mapsto \log(1 + e^{-z})$) or exponential ($z \mapsto e^{-z}$) loss using gradient flow (which is a continuous time analog of gradient descent) over a binary classification set $\{(x_i, y_i)\}_{i=1}^n \subseteq \mathbb{R}^d \times \{-1, 1\}$. Assume that there is a time $t_0$ where $L(\boldsymbol{\theta}(t_0)) < \frac{1}{n}$. Then, gradient flow converges in direction[1] to a first order stationary point (KKT point) of the following maximum-margin problem:*

$$\min_{\boldsymbol{\theta}} \frac{1}{2} \|\boldsymbol{\theta}\|^2 \ s.t \ \forall i \in [n] \ y_i \Phi(\boldsymbol{\theta}; x_i) \geq 1. \tag{1}$$

Since exploring privacy vulnerabilities is less interesting in networks with poor training accuracy, it is reasonable to assume that the training loss is reasonably small. Our paper specifically focuses on settings where, as stated in the above theorem, all training points are correctly classified. Therefore, throughout this paper, we assume that our target neural network has converged to a KKT point of Eq. (1). Formally, this implies the constraints captured in the following assumption:

---

[1]We say that gradient flow *converges in direction* to $\hat{\boldsymbol{\theta}}$ if $\lim_{t \to \infty} \frac{\boldsymbol{\theta}(t)}{\|\boldsymbol{\theta}(t)\|} = \frac{\hat{\boldsymbol{\theta}}}{\|\hat{\boldsymbol{\theta}}\|}$.

**Assumption 2.1.** *Let $\Phi(\boldsymbol{\theta}; \mathbf{x})$ be a 2-layer neural network, and let $m := \min_i |\Phi(\boldsymbol{\theta}; \mathbf{x}_i)| > 0$. We are given access to $\Phi(\boldsymbol{\theta}, \cdot)$, and we have full knowledge of the vector $\boldsymbol{\theta}$.[2] Moreover, we have that $\boldsymbol{\theta}$ satisfies the following KKT conditions of Eq. (1):*

$$\boldsymbol{\theta} = \sum_{i=1}^{n} \lambda_i y_i \nabla_{\boldsymbol{\theta}} \Phi(\boldsymbol{\theta}; \mathbf{x}_i), \tag{2}$$

$$\forall i \in [n], \quad y_i \Phi(\boldsymbol{\theta}; \mathbf{x}_i) \geq m > 0, \tag{3}$$

$$\lambda_1, \ldots, \lambda_n \geq 0, \tag{4}$$

$$\forall i \in [n], \quad \text{if } y_i \Phi(\boldsymbol{\theta}; \mathbf{x}_i) \neq m \text{ then } \lambda_i = 0. \tag{5}$$

We refer to the parameter $m$ as *the margin's value*, and we say that a set of points $A \subseteq \mathbb{R}^d$ *lies on the margin* if $\Phi(\boldsymbol{\theta}; \mathbf{x})$ equals the margin's value for all $\mathbf{x} \in A$. We stress that in general, the attacker does not have knowledge of the value of $m$. Nevertheless, it is still possible that the attacker might be able to either deduce this value or obtain it in some way, and even if they cannot, this merely results in a single additional hyperparameter that the attacker must accommodate for, which indicates that our proposed attacks can reveal unwanted information. Throughout this paper, we present several results which vary based on the information that we have on $m$.

## 3 ONE DIMENSIONAL INPUT

In this section, we consider univariate neural networks with ReLU activations. Such a network takes the form

$$x \mapsto \sum_{j=1}^{k} v_j \left[ w_j x + b_j \right]_+, \tag{6}$$

where $x \in \mathbb{R}$. Note that this computes a piece-wise linear function (in $x$), and its breakpoints (i.e. points where the function changes its linearity) are $\{-\frac{b_j}{w_j}\}_{j=1}^{k}$. Assume w.l.o.g. $-\frac{b_1}{w_1} < \ldots < -\frac{b_k}{w_k}$.

Throughout this section, we assume that the attacker has knowledge of the value of the margin, and that this value is 1 without loss of generality.

### 3.1 WARMING UP – THE CASE $n = k = 1$

It is easy to show that for the simple case of $n = k = 1$ there is a single possible solution, and thus the attacker can always recover the dataset:

**Theorem 3.1.** *Suppose that $\Phi(\boldsymbol{\theta}; \cdot)$ is a univariate neural network as in Eq. (6), and that Assumption 2.1 holds. Moreover, suppose that $n = k = 1$. Then, there exists a single solution $x$. Moreover, it can be easily recovered.*

*Proof.* Eq. (3) implies that $\Phi(\boldsymbol{\theta}; x_1)$ cannot be the zero function. By Eq. (5), $y_1 \Phi(\boldsymbol{\theta}; x_1) \neq 1$ implies that Eq. (2) equals zero which thus leads to a contradiction, and therefore we deduce that $y_1 \Phi(\boldsymbol{\theta}; x_1) = m = 1$ which implies that $\Phi(\boldsymbol{\theta}; x_1) \in \{-1, 1\}$. Since $n = k = 1$, we have $\Phi(\boldsymbol{\theta}; x_1) = v_1 [w_1 x_1 + b_1]_+$. This function equals 0 whenever the ReLU neuron is inactive and is necessarily not zero whenever the neuron is active, thus it has a non-zero slope, and it equals either $-1$ or 1 at a unique point which is necessarily $x_1$. □

While the above example is highly degenerate, it nevertheless highlights the danger and exemplifies the impact this theoretical tool may have in practice, and further motivates us to explore whether such vulnerabilities exist in more general settings.

---

[2]Many of our results or similar ones can be proven even with only partial access to the network's weights, however for the sake of simplicity we assume full knowledge of the weights.

## 3.2 THE GENERAL UNIVARIATE CASE

As we will see in this subsection, fully recovering the dataset in the more general univariate case as in the previous case is much more complicated – if at all possible. Nevertheless, we will show that under our assumptions, there is still some information on the training set that can be extracted.

Our analysis in the previous simple example relied on the observation that following from the KKT conditions, points whose value lies on the margin are potential candidates for being training points. However, it is unclear whether this holds in general, and what exactly is the portion of the points whose value lies on the margin that are also training points. Moreover, in the univariate case which we consider now, the neural network can either cross the margin with a non-zero slope, or have a zero slope on an interval where it equals the margin. In the former case, we have at most two points per each interval on which the network takes a linear form and crosses the margin, thus at most two points are added to the set of potential candidates; but in the latter case, there is a continuum of potential candidate points. However, a more careful analysis reveals that in both cases, there is a finite set of candidates which must contain a training point.

The following theorems each addresses a different case from the cases described above, and establishes the existence of a discrete set of points that must contain a training point. All proofs can be found in Appendix A.

**Theorem 3.2.** *Let $\Phi(\boldsymbol{\theta}; x)$ be a 2-layer univariate network satisfying Assumption 2.1. Let $\left[-\frac{b_{i-1}}{w_{i-1}}, -\frac{b_i}{w_i}\right]$ and $\left[-\frac{b_i}{w_i}, -\frac{b_{i+1}}{w_{i+1}}\right]$ be two adjacent intervals which none of them is constant on the margin. Then, there must be a training point in the interval $\left[-\frac{b_{i-1}}{w_{i-1}}, -\frac{b_{i+1}}{w_{i+1}}\right]$, and that training point must lie on the margin. In addition, the number of points lying on the margin in this interval is at most 4.*

The proof of the above theorem relies on the observation that for any three breaking points, two of them must belong to neurons with the same sign of the parameter $w$. If these two neurons are active on the same set of training points, then by Assumption 2.1, they merge into a single neuron, therefore there must exist some training point between them. Moreover, this training point must lie on the margin. Since each interval crosses the margin at most twice, the number of possible points lying on the margin is at most four.

Having presented our theorem for the case where the neural network is not constant on the margin, we now present our theorem for the complementary case where it is constant.

**Theorem 3.3.** *Let $\Phi(\boldsymbol{\theta}; x)$ be a 2-layer univariate network satisfying Assumption 2.1. In addition, assume the following:*

- *There is a neuron $c_1$ that is active on all the points in the training set.*

- *$\Phi(\boldsymbol{\theta}; x)$ is a local minimum of Eq. (1).*

- *$\Phi(\boldsymbol{\theta}; x)$ alternatingly lies on the margin on three adjacent intervals, i.e. it is constant on $\left[-\frac{b_{i-2}}{w_{i-2}}, -\frac{b_{i-1}}{w_{i-1}}\right]$ and on $\left[-\frac{b_i}{w_i}, -\frac{b_{i+1}}{w_{i+1}}\right]$ (but not in between) and lies on the margin, for some $i$.*

*Then, either $-\frac{b_{i-1}}{w_{i-1}}$ or $-\frac{b_i}{w_i}$ is a training point.*

If by contradiction neither $-\frac{b_{i-1}}{w_{i-1}}$ nor $-\frac{b_i}{w_i}$ is a training point, then we can construct a modified network with a slightly different breaking point $-\frac{b_i}{w_i} + \epsilon$ for any $\epsilon > 0$. We show that this new network has strictly smaller norm, yet it is still a feasible solution for Eq. (1) - A contradiction to $\Phi(\boldsymbol{\theta}, \cdot)$ having minimal norm.

We note that in terms of the structure of the function $\Phi(\boldsymbol{\theta}; \cdot)$, the above case analysis is exhaustive (excluding degenerate cases such as $\Phi(\boldsymbol{\theta}; \cdot)$ which consists of at most two different intervals, on which it is linear). This holds true since if the conditions in Thm. 3.3 do not hold, then this implies that $\Phi(\boldsymbol{\theta}; \cdot)$ does not lie on the margin in two adjacent intervals, hence the conditions for Thm. 3.2 must hold. We also remark that we have assumed that there is a neuron which is active on all the training data points, which typically makes sense in settings where the network is highly over-parameterized for example, but even if this assumption does not hold, then we can enforce it by

modifying our architecture to have a linear neuron with no activation function in the first hidden layer.

The next result demonstrates how our previous two theorems can be leveraged to construct a set of which at least a quarter of the instances are training points.

**Theorem 3.4.** *Let $\Phi : \mathbb{R} \to \mathbb{R}$ be a 2-layer homogeneous network satisfying Assumption 2.1. In addition, assume the following:*

- *There is a neuron $c_1$ that is active on all the points in the training set.*

- *$\Phi(\boldsymbol{\theta}; x)$ is a local minimum of Eq. (1).*

*Then, the following algorithm builds a finite set of which a constant fraction $p \geq \frac{1}{4}$ of the points are training points.*

---

**Algorithm 1** Build a finite set of candidates

---

1: $S \leftarrow \emptyset$
2: **for** $i = 1$ to $n - 2$ **do**
3:      $x \leftarrow -\frac{b_i}{w_i}$
4:      $y \leftarrow -\frac{b_{i+1}}{w_{i+1}}$
5:      $z \leftarrow -\frac{b_{i+2}}{w_{i+2}}$
6:      **if** both $[x, y]$ and $[y, z]$ do not lie on the margin **then**
7:          $S \leftarrow S \cup \{p : p \in [x, y] \cap p \text{ is on the margin}\} \cup \{p : p \in [y, z] \cap p \text{ is on the margin}\}$
8:      **end if**
9:      **if** $[x, y]$ lies on the margin **and** $i < n - 2$ **then**
10:         $t \leftarrow -\frac{b_{i+3}}{w_{i+3}}$
11:         **if** $[z, t]$ lies on the margin **then**
12:            $S \leftarrow S \cup \{y\} \cup \{z\}$
13:         **end if**
14:      **end if**
15: **end for**

---

The above algorithm essentially iterates over the linear intervals of the network, and uses either Thm. 3.2 or Thm. 3.3 based on the structure of $\Phi(\boldsymbol{\theta}; \cdot)$ to add a constant number of candidate points, until the final set of points is constructed. We point out that we have assumed that $\boldsymbol{\theta}$ is a local minimum of Eq. (1) rather than just a critical point. It is known that in general, not all critical points of Eq. (1) are also local minima, and that gradient flow may converge to a critical point which is not a local minimum (see Safran et al. (2022, Example 1)), but it is not clear what is the 'typical' behavior of gradient flow in this context. We also remark that despite our requirement to have full knowledge of $\boldsymbol{\theta}$, the above results can also be implemented with partial knowledge of $\boldsymbol{\theta}$.[3] In any case, we leave the exploration of other privacy related questions on relaxations of our assumptions for future work.

## 4   HIGH DIMENSIONAL INPUT

Having discussed the one-dimensional setting, we now investigate the case $\mathbf{x} \in \mathbb{R}^d$ where $d$ is large. In this case, it is not obvious how to reconstruct the training data using an approach which is similar to the previous section: even if one can identify a $(d - 1)$-dimensional manifold (which corresponds to domain points that lie on the margin) in which the data is contained, there is still a continuum of potential candidates. For this reason, we instead investigate a different variant of privacy vulnerability, called *membership inference* queries: Given a point $\mathbf{x} \in \mathbb{R}^d$ which is either a random point from the training set or a freshly sampled test point, sampled from the same distribution used to generate the training set – can the attacker tell how $\mathbf{x}$ was generated with high probability?

---

[3]For example, if we have access to $\Phi(\boldsymbol{\theta}; \cdot)$ and only the breakpoints where the network changes its linearity are known, we can still interpolate and compute the points which cross the margin.

In very high dimensional settings, under many commonly used data distributions, we have that the dataset is almost orthogonal with high probability. We exploit this property to show that also with high probability over drawing the training set, all the points in the training set will lie on the margin. On the other hand, if we draw a new data point from the same distribution, the neural network will output a target value which is typically much smaller than the margin. These key observations will allow us to make the distinction between training points and test points, effectively answering membership inference queries.

**Remark 4.1** (Black box attacks)**.** *We note that since our results in this section are only based on querying the value of $\Phi(\boldsymbol{\theta}; \cdot)$, the attacker need not know $\boldsymbol{\theta}$ to successfully execute the membership inference attack, and therefore the attack can also be applied in the black box model.*

We now formally state our assumptions on the underlying distribution $\mathcal{D}$ which generates the dataset:

**Assumption 4.1.** *The following hold for some $\tau > 0$.*

1. *For $\mathbf{x}_1, \mathbf{x}_2 \sim \mathcal{D}, \quad \Pr[n \cdot |\mathbf{x}_1^\top \mathbf{x}_2| \leq o(d)] \geq 1 - \frac{\tau}{n^2}.$*

2. *For $\mathbf{x} \sim \mathcal{D}, \quad p\left[\|\mathbf{x}\|^2 \geq \Omega(d)\right] \geq 1 - \frac{\tau}{n}.$*

*where $n$ is the size of the training set.*

Note that we do not make any assumptions on the labels of the data, and therefore our results hold for all possible labeling on the data. We also point out that even though this assumption may seem somewhat restrictive at a first glance, it can be expected to hold for continuous distributions in sufficiently large dimensions, and when the sample size is modest. We also prove that our assumption is satisfied by several rather standard data distributions. This includes (but is not limited to) the following concrete examples:

1. The uniform distribution over the sphere $\sqrt{d} \cdot \mathbb{S}^{d-1}$, where $n = o\left(\frac{\sqrt{d}}{\log d}\right)$ and $\tau = o_d(1)$.

2. The normal distribution $\mathcal{N}(\boldsymbol{\mu}, I)$ with mean $\boldsymbol{\mu}$, where $\|\boldsymbol{\mu}\|^2 = o(d)$, and where $n = \frac{o(d)}{\|\boldsymbol{\mu}\|^2 + d^\epsilon}$ for some $\frac{1}{2} < \epsilon < 1$ and $\tau = o_d(1)$.

3. Mixture of $k$ Gaussians with means $\boldsymbol{\mu}^{(1)}, \ldots, \boldsymbol{\mu}^{(k)}$, where $\|\boldsymbol{\mu}^{(1)}\|^2, \ldots, \|\boldsymbol{\mu}^{(k)}\|^2 = o(d)$, identity covariance matrices, $n = \frac{o(d)}{\max\{\|\boldsymbol{\mu}^{(i)}\|^2\}_{i=1}^k + d^\epsilon}$ for some $\frac{1}{2} < \epsilon < 1$, and $\tau = o_d(1)$.

The first two examples are rather standard in the literature, whereas the last example is somewhat more complex, but is meant to exemplify a setting where our proposed attacks can be executed in the statistically learnable case. For a more formal discussion about the statistically learnable case, we refer the reader to Appendix C. For proofs that these distributions satisfy Assumption 4.1, we refer the reader to Appendix D.

Before we continue, we will introduce some further notation to be used throughout this section. Recall that $m > 0$ denotes the value of the network's margin, and define $\delta := \max_{i \neq j} \left\{|\mathbf{x}_i^\top \mathbf{x}_j|\right\}$ and $\Delta := \min_{i \in [n]} \left\{\|x_i\|^2\right\}$. Note that by Assumption 4.1 and by the union bound, we have that $n \cdot \delta = o(\Delta)$ with probability at least $1 - 2\tau$.

Given a point $\mathbf{x} \in \mathbb{R}^d$, we would like to know whether $\mathbf{x}$ was in the training set, or if it was generated from the same distribution that generated the training set. As previously discussed, our strategy is to calculate the value of $|\Phi(\boldsymbol{\theta}; \mathbf{x})|$. We expect to see larger values that are closer to the margin when $\mathbf{x}$ is in the training set, and smaller values when it is not. Formalizing this idea, the following theorem is used to determine w.h.p. whether a given point $\mathbf{x} \in \mathbb{R}^d$ is in fact a training point, or a test point which was freshly sampled from $\mathcal{D}$.

**Theorem 4.2.** *Let $\mathcal{D}$ be a distribution on $\mathbb{R}^d$ that satisfies Assumption 4.1. Let $\mathbf{x} \in \mathbb{R}^d$ and let $\Phi(\boldsymbol{\theta}; \cdot)$ be a 2-layer neural network satisfying Assumption 2.1. Then the following hold:*

- *With probability at least $1 - 2\tau$ over the choice of the training set, if $\mathbf{x}$ is a training point then $|\Phi(\boldsymbol{\theta}; \mathbf{x})| = m$.*

- *If $\mathbf{x} \sim \mathcal{D}$ then with probability $1 - 2\tau$ over the sampling of $\mathbf{x}$ and the sampling of the training data, $|\Phi(\boldsymbol{\theta}; \mathbf{x})| = O(\frac{n \cdot m \cdot \delta}{\Delta}) = o_d(m).$*

This theorem gives us a useful tool to perform membership inference attacks. Given a point $\mathbf{x} \in \mathbb{R}^d$, run $\mathbf{x}$ through the neural network, and consider the output $\Phi(\boldsymbol{\theta}; \mathbf{x})$. If $|\Phi(\boldsymbol{\theta}; \mathbf{x})| = m$, then $\mathbf{x}$ is in the training set, and if $|\Phi(\boldsymbol{\theta}; \mathbf{x})| = o_d(m)$, then $\mathbf{x}$ is not in the training set (with high probability).

The intuition behind the proof of the theorem can be explained as follows: Using Assumption 2.1, we show that the value $|\Phi(\boldsymbol{\theta}; \mathbf{x})|$ can be expressed as a weighted combination of $\{\mathbf{x}_i^\top \mathbf{x}\}_{i=1}^n$ (where $\{\mathbf{x}_i\}_{i=1}^n$ are the training points). Using Assumption 4.1, we know that if $\mathbf{x}$ is in the training set, then $\mathbf{x} = \mathbf{x}_k$ for some $k \in [n]$ and $\|\mathbf{x}_k\|^2$ must be large, while $\mathbf{x}_j^\top \mathbf{x}_k$ is small for all $j \neq k$, and therefore the weighted combination is large. On the other hand, when $\mathbf{x} \sim \mathcal{D}$, then with high probability it is "nearly orthogonal" to all training points, meaning that $\mathbf{x}_j^\top \mathbf{x}$ is small for all $j = 1, \ldots, n$, and thus the weighted combination is small. For the complete proof of the theorem, we refer the reader to Appendix B.

Having presented our main tool in this section, we now turn to discuss several particular use cases, based on the amount of knowledge known to the attacker. Similarly to the previous section, we first assume that the value of the margin is known to the attacker. However, since an attacker cannot deduce the value of the margin in general, we also provide examples where membership inference questions can be answered without this knowledge.

## 4.1 Example use cases of Thm. 4.2

In all of the following cases, let $\Phi(\boldsymbol{\theta}; \mathbf{x})$ be a 2-layer neural network satisfying Assumption 2.1, and let $\mathcal{D}$ be a distribution that satisfies Assumption 4.1, so as to satisfy the assumptions in Thm. 4.2.

We begin with the simplest case, where the value of the margin is known to the attacker.

**Corollary 4.3** (Known margin value). *Let $\mathbf{x} \in \mathbb{R}^d$, assume that $d$ is sufficiently large, and further assume that we know the value of the margin $m$. Then, w.h.p. over the randomness in sampling the training set from $\mathcal{D}$, we have that:*

- *If $\mathbf{x}$ is in the training set then $|\Phi(\boldsymbol{\theta}; \mathbf{x})| = m$.*

- *If $\mathbf{x} \sim \mathcal{D}$ is a fresh example, then w.h.p. over the randomness in sampling $\mathbf{x}$, $|\Phi(\boldsymbol{\theta}; \mathbf{x})| < \frac{m}{2}$.*

*Proof.* From Thm. 4.2 we know that w.h.p. over the choice of the training set we have that if $\mathbf{x}$ is in the training set then $|\Phi(\boldsymbol{\theta}; \mathbf{x})| = m$ and if $\mathbf{x} \sim \mathcal{D}$ then w.h.p.

$$|\Phi(\boldsymbol{\theta}; \mathbf{x})| \leq O\left(\frac{n \cdot m \cdot \delta}{\Delta}\right) = m \cdot O\left(\frac{n \cdot \delta}{\Delta}\right) < \frac{m}{2},$$

where in the last inequality we used the fact that $O(\frac{n \cdot \delta}{\Delta}) = o_d(1)$. $\qquad\square$

Thus, by the above, if the margin's value $m$ is known to the attacker, they can simply compute $|\Phi(\boldsymbol{\theta}; \mathbf{x})|$ and return that $\mathbf{x}$ is in the training set if and only if $|\Phi(\boldsymbol{\theta}; \mathbf{x})| \approx m$.

As previously discussed, in general, the value of the margin is not known to the attacker. Nevertheless, under different assumptions, the attacker can still execute a successful membership inference attack.

**Corollary 4.4** (Leaked data point). *Let $k$ be a constant (independent of $d$), let $\mathbf{z}_1, \ldots, \mathbf{z}_k \sim \mathcal{D}$ be $k$ points, and assume we know that at least one point in this set is in the training set. Let $\alpha = \max_{1 \leq i \leq k} \{|\Phi(\boldsymbol{\theta}; \mathbf{z}_i)|\}$, then w.h.p. over the choice of the training set, we have for all $i \in [k]$:*

- *If $\mathbf{z}_i$ is in the training set then $|\Phi(\boldsymbol{\theta}; \mathbf{z}_i)| = \alpha$.*

- *If $\mathbf{z}_i \sim \mathcal{D}$ then w.h.p. (over sampling $\mathbf{z}_i$) $|\Phi(\boldsymbol{\theta}; \mathbf{z}_i)| < \frac{\alpha}{2}$.*

*Proof.* W.l.o.g. let $\mathbf{z}_1$ be in the training set. Using Thm. 4.2 and the union bound over $\mathbf{z}_1, \ldots, \mathbf{z}_k$, we have $|\Phi(\boldsymbol{\theta}, \mathbf{z}_i)| \leq m$ for all $i$ with probability at least $1 - 2k\tau = 1 - o_d(1)$, so in particular $\alpha \leq m$. On the other hand, using Thm. 4.2 again, we have that with probability at least $1 - \tau = 1 - o_d(1)$ we have that $|\Phi(\boldsymbol{\theta}, \mathbf{z}_1)| = m$, so $m \leq \alpha$. So we have that w.h.p. $m = \alpha$. Now we complete the proof by using Corollary 4.3. $\qquad\square$

The above corollary implies that even if the attacker has no knowledge of the value of the margin, but has knowledge that at least one element in a set of size $k$ is in the training set, then this value must achieve the maximal prediction value in absolute value among the set. This allows the attacker to deduce the margin value by computing $\max_i |\Phi(\boldsymbol{\theta}; \mathbf{z}_i)|$. Thereafter, the attacker can continue in the same manner as in Corollary 4.3.

One might argue that even the previous assumptions are somewhat restrictive, since they require that at least one training point is leaked a priori. The following corollary makes some additional assumptions on the underlying distribution and that the margin value is bounded rather than known, which is much milder than in the previous result.

**Corollary 4.5** (Bounded margin). *Let $\mathcal{D}$ be a distribution that satisfies the following slightly stronger version of Assumption 4.1:*

*Let $\tau > 0$.*

- *For $\mathbf{x}, \mathbf{y} \sim \mathcal{D}$, $n \cdot |\mathbf{x}^\top \mathbf{y}| = o\left(\frac{d}{t(d)}\right)$ for some function $t(d)$ with probability at least $1 - \frac{\tau}{n^2}$.*

- *For $\mathbf{x} \sim \mathcal{D}$, $\|\mathbf{x}\|^2 = \Omega(d)$ with probability at least $1 - \frac{\tau}{n}$.*

*Furthermore, let $\mathbf{x} \sim \mathcal{D}$ and suppose that $C < m < t(d)$ for some constant $C$. Then the following holds:*

- *W.p. at least $1 - \tau$ over the training set, if $\mathbf{x}$ is in the training set then $|\Phi(\boldsymbol{\theta}; \mathbf{x})| > C$.*

- *If $\mathbf{x} \sim \mathcal{D}$ then w.p. at least $1 - 2\tau$ over the training set and $\mathbf{x}$, $|\Phi(\boldsymbol{\theta}; \mathbf{x})| < o_d(1)$.*

*Proof.* Assume that $\mathbf{x}$ is in the training set. From Thm. 4.2 we know that $|\Phi(\boldsymbol{\theta}; \mathbf{x})| = m > C$ with probability at least $1 - \tau$. Assume that $\mathbf{x}$ is not in the training set. From Thm. 4.2 and our stronger assumption on $\mathcal{D}$ we know that

$$|\Phi(\boldsymbol{\theta}; \mathbf{x})| = O\left(\frac{n \cdot \delta \cdot m}{\Delta}\right) \leq O\left(o\left(\frac{d}{t(d)}\right) \cdot \frac{m}{\Delta}\right) = O\left(o\left(\frac{d}{t(d)}\right) \cdot \frac{m}{d}\right) = o_d(1).$$

with probability at least $1 - 2\tau$. $\qquad\square$

This corollary implies the following: for $\mathbf{x} \in \mathbb{R}^d$, let us compute $|\Phi(\boldsymbol{\theta}; \mathbf{x})|$. If $\mathbf{x}$ is not in the training set, then w.h.p. we get a number which is smaller than $C$, and if $\mathbf{x}$ is in the training set, w.h.p we get a number which is larger than $C$.

**Remark 4.6** (On the lower and upper bounds of the margin). *We argue that the lower and upper bound assumptions on the margin that we use above are mild. This follows from the fact that if we assume an exponential or logistic loss function (which is a standard assumption in this setting), then the gradient is exponentially small in the margin. Hence, if the margin is even just polylogarithmic in $d$, then making further progress with training is extremely inefficient. Conversely, if the margin is too small, then this implies that the loss over points that lie on the margin is large, which indicates that the network had stopped training very early. For more formal arguments justifying this assumption, we refer the reader to Remarks B.5 and B.6.*

## 5    AN EXPERIMENT FOR INTERMEDIATE VALUES OF $d$

Thus far, our theory addressed the one-dimensional case, as well as the high-dimensional case where the input's dimension is much larger than the training set size. This naturally raises the question of what happens in between these two regimes.

Exploring this question empirically, in this section, we conducted a few experiments focusing on the membership inference problem, and observed that while our theoretical results' assumptions do not necessarily hold, their implications are nevertheless still valid. We sampled training and test sets (both i.i.d.) uniformly from the scaled hypersphere, trained a 2-layer neural network until reaching an approximate KKT point, and examined the network's predictions on both the training and the test sets in comparison to the margin. Our code is available in the supplementary material.

More specifically, we conducted all our simulations using the following settings:

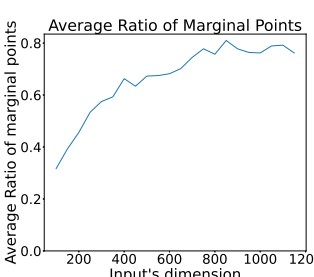 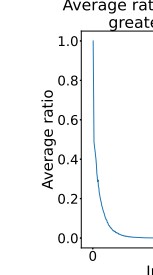 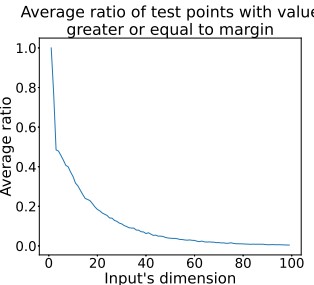

(a) The percentage of training points that lie on the margin (up to a slack of 10%) increases as the dimension increases.

(b) The percentage of test points that lie on or above the margin drops to zero for sufficiently large input dimensions, much earlier than what our theory predicts.

(c) A zoom in to the smaller values. The percentage of test points that lie on or above the margin decreases rapidly as the dimension increases.

Figure 1: The relative values of training and test points compared to the value of the margin, where every point in the above graphs was averaged over 50 instantiations.

- **Architecture:** We focused on 2-layer ReLU networks, where the hidden layer has 10,000 neurons. The neurons in the hidden layer each have a bias term while the second layer does not, thus making the network homogeneous.

- **Range of the input dimension:** We tested $d$ for various values in the range between 1 and 1000. This range includes values of $d$ where it is much larger than the training set as in our theoretical results, but also includes more moderate values of $d$ where our assumptions do not necessarily hold.

- **Data generation:** All points were sampled uniformly and i.i.d. from $\sqrt{d} \cdot \mathbb{S}^{d-1}$. The training set contains 20 instances, since this small size ensures that Assumption 4.1 holds for the larger values of $d$ that we tested. The test set contains 5,000 instances.

- **Training:** In order to converge faster to an approximate KKT point, we used a small initialization scheme as was done in Haim et al. (2022).

Our experiment focused on studying two objectives. The first studies how many training points lie on the margin as a function of the dimension $d$,[4] and the second studies how many test points that were sampled from the same distribution as the training set lie on or above the margin.

Our results demonstrate that network outputs can serve as effective tools for privacy attacks across a broader range of input dimensions, suggesting wider applicability of our theory. Specifically, Fig. 1a shows that as input dimensions increase, more training points lie on the margin, indicating a higher probability of this occurrence. Similarly, Fig. 1b and Fig. 1c reveal that the number of test points lying on or above the margin decreases with higher dimensions, implying a reduced likelihood of test points from the same distribution doing so. Notably, these findings align with our theory and extend to much smaller dimensions than predicted. For instance, while Thm. 4.2 suggests a minimum dimension of $d = n^2 = 400$[5] for a training set of size 20, our experiments show that nearly all test points fall below the margin even at $d = 100$, and about 80% do so at $d = 20$, highlighting the potential for membership inference attacks at much lower dimensions.

Following our empirical findings, we conclude that our theory is expected to hold more generally, and that the magnitude of the output of the neural network on a data instance can provably reveal whether it is a training point or a test point with high success rates. This is in line with many empirical findings (see Hu et al. (2022b)), and provides a theoretical explanation for this phenomenon.

---

[4]It is noteworthy that a similar experiment was conducted in Vardi et al. (2022b), albeit under a different context where the adversarial robustness of the neural network is studied.

[5]This is because of the fact that under the assumption $n = \sqrt{d}$, we have w.h.p. that $n \cdot |\mathbf{x}_1^\top \mathbf{x}_2| = \Theta(d)$, so Assumption 4.1 is very unlikely to hold for values of $d$ that are smaller than that.

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

## A  PROOFS FROM SUBSECTION 3.2

We start by stating a few notations: Denote by $\sigma'_j$ the subgradient of $\left[\mathbf{w}_j^\top \mathbf{x} + b_j\right]_+$. If $\mathbf{w}_j^\top \mathbf{x} + b_j \neq 0$ then $\sigma'_j$ is well defined, and if $\mathbf{w}_j^\top \mathbf{x} + b_j = 0$ then $\sigma'_j \in [0, 1]$. In any case, $\sigma'_j \geq 0$. For a training point $\mathbf{x}_i$, denote by $\sigma'_{i,j}$ the subgradient of $\left[\mathbf{w}_j^\top \mathbf{x}_i + b_j\right]_+$.

For all $j \in [k]$ that the partial derivatives of our 2-layer homogeneous neural network are given by

$$\frac{\partial}{\partial v_j}\Phi(\boldsymbol{\theta}; \mathbf{x}) = \left[\mathbf{w}_j^\top \mathbf{x} + b_j\right]_+,$$

$$\frac{\partial}{\partial \mathbf{w}_j}\Phi(\boldsymbol{\theta}; \mathbf{x}) = v_j x \sigma'_j,$$

$$\frac{\partial}{\partial b_j}\Phi(\boldsymbol{\theta}; \mathbf{x}) = v_j \sigma'_j.$$

Combining the above with the KKT conditions, we arrive at

$$v_j = \sum_{i=1}^n \lambda_i y_i \left[\mathbf{w}_j^\top \mathbf{x}_i + b_j\right]_+, \tag{7}$$

$$\mathbf{w}_j = v_j \sum_{i=1}^n \lambda_i y_i \mathbf{x}_i \sigma'_{i,j}, \tag{8}$$

$$b_j = v_j \sum_{i=1}^n \lambda_i y_i \sigma'_{i,j}, \tag{9}$$

for all $j \in [k]$.

**Lemma A.1.** *Let $\phi$ be a 2-layer homogeneous network that satisfy the KKT conditions. Let $x_l < x_{l+1}$ be 2 adjacent marginal training points. The number of breaking points in the interval $[x_l, x_{l+1}]$ is at most 2, i.e. $|\{-\frac{b_j}{w_j} : \ x_l \leq -\frac{b_j}{w_j} \leq x_{l+1}\}| \leq 2$. Moreover, if there are 2 breaking points, the neurons forming the breaking points must have different signs.*

*Proof.* Let $c_{j_1}(x) = v_{j_1}[w_{j_1}x + b_{j_1}]_+$ and $c_{j_2}(x) = v_{j_2}[w_{j_2}x + b_{j_2}]_+$ be 2 neurons with $w_{j_1} < 0$ and $w_{j_2} < 0$ such that their breaking points are between $x_l$ and $x_{l+1}$. Both $c_{j_1}$ and $c_{j_2}$ are determined by all the training points that are smaller than $x_{l+1}$. let us look at their breaking point $-\frac{b_l}{w_l}$ and $-\frac{b_{l+1}}{w_{l+1}}$. From Eq. (8) and Eq. (9) we get that

$$-\frac{b_{j_1}}{w_{j_1}} = -\frac{v_{j_1}\sum_{i=1}^l \lambda_i y_i}{v_{j_1}\sum_{i=1}^l \lambda_i y_i x_i} = -\frac{\sum_{i=1}^l \lambda_i y_i}{\sum_{i=1}^l \lambda_i y_i x_i} = -\frac{v_{j_2}\sum_{i=1}^l \lambda_i y_i}{v_{j_2}\sum_{i=1}^l \lambda_i y_i x_i} = -\frac{b_{j_2}}{w_{j_2}}$$

This means the neurons have the same breaking point and are active on the same region, which means they are the same neuron.

The same argument can be made to show that if $w_l > 0$ and $w_{l+1} > 0$ the neurons have the same breaking point.

We conclude that in this interval we can have at most one neuron with $w > 0$ and at most one neuron with $w < 0$ with breaking points in the interval $[x_l, x_{l+1}]$. $\qquad\square$

**Lemma A.2.** *Let $x_1 < x_2 < \cdots < x_n$ be the training points on the margin and $\phi(x; \theta)$ be a 2-layers NN. If The network $\phi(x; \theta)$ satisfies the KKT conditions, and is not constant in any interval, then the number of times it crosses the margin is at most $6n$.*

*Proof.* between each $x_l$, $x_{l+1}$ there are at most 2 breaking points, i.e the networks crosses the margin at most 6 times in the interval $[x_l, x_{l+1}]$ (3 times the the margin $y = 1$ and 3 times the margin $y = -1$). Before the point $x_1$ and after the point $x_n$ the network crosses the line at most 6 times in each interval. So if we sum up all the crosses we get that the network crosses the margin at most $6 \cdot (n-2) + 12 = 6n$ $\qquad\square$

Figure 2: The blue network is a network which the breaking point is not a training point. The dotted-red network has smaller norm.

*Proof Of Thm. 3.2.* Assume towards contradiction that there are no training points in the interval $[-\frac{b_{i-1}}{w_{i-1}}, -\frac{b_{i+1}}{w_{i+1}}]$. Since there are 3 breaking points, two of the neurons must have the same sign. W.l.o.g $sgn(w_{i-1}) = sgn(w_i)$ (all other cases are similar). Since there are no marginal training data in $[-\frac{b_{i-1}}{w_{i-1}}, -\frac{b_{i+1}}{w_{i+1}}]$, they are active on the same set of training points, which means by Eq. (8) and Eq. (9) that $-\frac{b_{i-1}}{w_{i-1}} = -\frac{b_i}{w_i}$.

Each interval crosses the margin at most twice, so the number points lying on the margin is at most 4. $\qquad\square$

*Proof Of Thm. 3.3.* This proof follows the same logic as the proof of Lemma A.6 in Kornowski et al. (2023).

Assume towards contradiction that neither $-\frac{b_i}{w_i}$ nor $-\frac{b_{i+1}}{w_{i+1}}$ are in the training set, if $x \in [-\frac{b_i}{w_i}, -\frac{b_{i+1}}{w_{i+1}}]$ then $x \in (-\frac{b_i}{w_i}, -\frac{b_{i+1}}{w_{i+1}})$.

Note that $sgn(w_{i-1}) = -sgn(w_i)$ because there is no training point in the interval $(-\frac{b_i}{w_i}, -\frac{b_{i+1}}{w_{i+1}})$ so by A.1 they must have different signs.

Also note that there must be a training point either in $[-\frac{b_{i-2}}{w_{i-2}}, -\frac{b_{i-1}}{w_{i-1}}]$ or in $[-\frac{b_i}{w_i}, -\frac{b_{i+1}}{w_{i+1}}]$ (or in both). If it is not the case there are at least 3 breaking points between to training data points, contradiction to A.1.

**CASE 1**: $v_i^2 + \frac{v_i w_i v_{i-1}}{w_{i-1}} + \frac{b_i}{1-\delta}\left(\frac{w_i b_{i-1}}{w_{i-1}} - b_i\right) - \frac{w_1 v_i w_i}{v_1} - \frac{b_1 b_{i-1} v_i w_i}{v_1 w_{i-1}} > 0$
Define the following neural network:

$$\phi(\theta_\delta, x) := \sum_{j \in [n] \setminus \{i-1, i, 1\}} v_j \left[w_j \cdot x + b_j\right]_+ + \left(1 - \delta \frac{v_i w_i}{v_{i-1} w_{i-1}}\right) v_{i-1} \left[w_{i-1} x + b_{i-1}\right]_+ +$$

$$(1-\delta) v_i \left[w_i x + b_i - \frac{\delta}{1-\delta}\left(\frac{w_i b_{i-1}}{w_{i-1}} - b_i\right)\right]_+ +$$

$$v_1 \left[\left(w_1 + \delta \frac{v_i w_i}{v_1}\right) x + \left(b_1 + \delta \frac{v_i w_i b_{i-1}}{v_1 w_{i-1}}\right)\right]_+$$

For small enough $\delta$, the new breaking points do not cross any training point so for any training point $x_j$ we have that $\phi(\theta, x_j) = \phi(\theta_\delta, x_j)$ and in particular $\phi(\theta_\delta, x)$ satisfies the margin condition

for each training point $x_j$. Also note that $\|\phi(\theta, x) - \phi(\theta_\delta, x)\|^2 \to 0$ as $\delta \to 0$. Let us compute $\|\phi(\theta_\delta, x)\|^2$:

$$\|\phi(\theta_\delta, x)\|^2 = \sum_{j \in [n] \setminus \{i-1, i, 1\}} (v_j^2 + w_j^2 + b_j^2) + \left(1 - \delta \frac{v_i w_i}{v_{i-1} w_{i-1}}\right)^2 v_{i-1}^2 + w_{i-1}^2 + b_{i-1}^2 +$$

$$(1-\delta)^2 v_i^2 + w_i^2 + \left(b_i - \frac{\delta}{1-\delta}\left(\frac{w_i b_{i-1}}{w_{i-1}} - b_i\right)\right)^2 +$$

$$v_1^2 + \left(w_1 + \delta \frac{v_i w_i}{v_1}\right)^2 + \left(b_1 + \delta \frac{v_i w_i b_{i-1}}{v_1 w_{i-1}}\right)^2 =$$

$$\|\phi(\theta, x)\|^2 - 2\delta \left(v_i^2 + \frac{v_i w_i v_{i-1}}{w_{i-1}} + \frac{b_i}{1-\delta}\left(\frac{w_i b_{i-1}}{w_{i-1}} - b_i\right) - \frac{w_1 v_i w_i}{v_1} - \frac{b_1 b_{i-1} v_i w_i}{v_1 w_{i-1}}\right) + O(\delta^2)$$

$$< \|\phi(\theta, x)\|^2$$

**CASE 2**: $v_i^2 + \frac{v_i w_i v_{i-1}}{w_{i-1}} + \frac{b_i}{1-\delta}\left(\frac{w_i b_{i-1}}{w_{i-1}} - b_i\right) - \frac{w_1 v_i w_i}{v_1} - \frac{b_1 b_{i-1} v_i w_i}{v_1 w_{i-1}} < 0$

Define the following neural network:

$$\phi(\theta_\delta, x) := \sum_{j \in [n] \setminus \{i-1, i, 1\}} v_j \left[w_j \cdot x + b_j\right]_+ + \left(1 + \delta \frac{v_i w_i}{v_{i-1} w_{i-1}}\right) v_{i-1} \left[w_{i-1} x + b_{i-1}\right]_+ +$$

$$(1 + \delta) v_i \left[w_i x + b_i + \frac{\delta}{1+\delta}\left(\frac{w_i b_{i-1}}{w_{i-1}} - b_i\right)\right]_+ +$$

$$v_1 \left[\left(w_1 - \delta \frac{v_i w_i}{v_1}\right) x + \left(b_1 - \delta \frac{v_i w_i b_{i-1}}{v_1 w_{i-1}}\right)\right]_+$$

The norm $\|\phi(\theta_\delta, x)\|^2$ is:

$$\|\phi(\theta_\delta, x)\|^2 = \sum_{j \in [n] \setminus \{i-1, i, 1\}} (v_j^2 + w_j^2 + b_j^2) + \left(1 + \delta \frac{v_i w_i}{v_{i-1} w_{i-1}}\right)^2 v_{i-1}^2 + w_{i-1}^2 + b_{i-1}^2 +$$

$$(1+\delta)^2 v_i^2 + w_i^2 + \left(b_i + \frac{\delta}{1-\delta}\left(\frac{w_i b_{i-1}}{w_{i-1}} - b_i\right)\right)^2 +$$

$$v_1^2 + \left(w_1 - \delta \frac{v_i w_i}{v_1}\right)^2 + \left(b_1 - \delta \frac{v_i w_i b_{i-1}}{v_1 w_{i-1}}\right)^2 =$$

$$\|\phi(\theta, x)\|^2 - 2\delta \left(-v_i^2 - \frac{v_i w_i v_{i-1}}{w_{i-1}} - \frac{b_i}{1-\delta}\left(\frac{w_i b_{i-1}}{w_{i-1}} - b_i\right) + \frac{w_1 v_i w_i}{v_1} + \frac{b_1 b_{i-1} v_i w_i}{v_1 w_{i-1}}\right) + O(\delta^2)$$

$$< \|\phi(\theta, x)\|^2$$

**CASE 3**: $v_i^2 + \frac{v_i w_i v_{i-1}}{w_{i-1}} + \frac{b_i}{1-\delta}\left(\frac{w_i b_{i-1}}{w_{i-1}} - b_i\right) - \frac{w_1 v_i w_i}{v_1} - \frac{b_1 b_{i-1} v_i w_i}{v_1 w_{i-1}} = 0$

In this case, define the following neural network:

$$\phi(\theta_\delta, x) := \sum_{j \in [n] \setminus \{i-1, i, 1\}} v_j \left[w_j \cdot x + b_j\right]_+ +$$

$$(1 - \delta) v_{i-1} \left[w_{i-1} x + b_{i-1} - \frac{\delta}{1-\delta}\left(\frac{w_{i-1} b_i}{w_i} - b_{i-1}\right)\right]_+ +$$

$$\left(1 - \delta \frac{v_{i-1} w_{i-1}}{v_i w_i}\right) v_i \left[w_i x + b_i\right]_+ +$$

$$v_1 \left[\left(w_1 + \delta \frac{v_{i-1} w_{i-1}}{v_1}\right) x + b_1 + \delta \frac{v_{i-1} w_{i-1} b_i}{v_1 w_i}\right]_+$$

Before computing the norm, let us note a two observations:

1. By assumption, $v_i w_i = -v_{i-1} w_{i-1}$ and hence $\frac{v_i}{w_{i-1}} = -\frac{v_{i-1}}{w_i}$,

2. By definition of case 3, $v_i^2 + \frac{v_i w_i v_{i-1}}{w_{i-1}} + \frac{b_i}{1-\delta}\left(\frac{w_i b_{i-1}}{w_{i-1}} - b_i\right) - \frac{w_1 v_i w_i}{v_1} - \frac{b_1 b_{i-1} v_i w_i}{v_1 w_{i-1}} = 0$,

Now let us compute the norm:

$$\|\phi(\theta_\delta, x)\|^2 = \sum_{j \in [n] \setminus \{i-1, i, 1\}} (v_j^2 + w_j^2 + b_j^2) + (1-\delta)^2 v_{i-1}^2 + w_{i-1}^2 + \left(b_{i-1} - \frac{\delta}{1-\delta}(\frac{w_{i-1} b_i}{w_i} - b_{i-1})\right)^2 +$$

$$\left(1 - \delta\frac{v_{i-1} w_{i-1}}{v_i w_i}\right)^2 v_i^2 + w_i^2 + b_i^2 +$$

$$\left(w_1 + \delta\frac{v_{i-1} w_{i-1}}{v_1}\right)^2 + \left(b_1 + \delta\frac{v_{i-1} w_{i-1} b_i}{v_1 w_i}\right)^2 =$$

$$\|\phi(\theta, x)\|^2 - 2\delta\left(v_{i-1}^2 + \frac{b_{i-1}}{1-\delta}(\frac{w_{i-1} b_i}{w_i} - b_{i-1}) + \frac{v_{i-1} w_{i-1} v_i}{w_i} - \frac{w_1 v_{i-1} w_{i-1}}{v_1} - \frac{b_1 b_i v_{i-1} w_{i-1}}{v_1 w_i}\right) + O(\delta^2)$$

We need to show that

$$v_{i-1}^2 + \frac{b_{i-1}}{1-\delta}\left(\frac{w_{i-1} b_i}{w_i} - b_{i-1}\right) + \frac{v_{i-1} w_{i-1} v_i}{w_i} - \frac{w_1 v_{i-1} w_{i-1}}{v_1} - \frac{b_1 b_i v_{i-1} w_{i-1}}{v_1 w_i} \neq 0$$

(if $v_{i-1}^2 + \frac{b_{i-1}}{1-\delta}(\frac{w_{i-1} b_i}{w_i} - b_{i-1}) + \frac{v_{i-1} w_{i-1} v_i}{w_i} - \frac{w_1 v_{i-1} w_{i-1}}{v_1} - \frac{b_1 b_i v_{i-1} w_{i-1}}{v_1 w_i} < 0$ then, as in the previous cases, we change every $\delta$ to $-\delta$ and every $-\delta$ to $\delta$).
By observation 1 we know that:

$$\frac{v_{i-1} w_{i-1} v_i}{w_i} = -\frac{v_i w_i v_i}{w_i} = -v_i^2 \tag{10}$$

$$\frac{v_i w_i v_{i-1}}{w_{i-1}} = -\frac{v_{i-1} w_{i-1} v_i}{w_{i-1}} = -v_{i-1}^2 \tag{11}$$

Combine this with observation 2 we get:

$$v_i^2 + \frac{v_i w_i v_{i-1}}{w_{i-1}} + \frac{b_i}{1-\delta}(\frac{w_i b_{i-1}}{w_{i-1}} - b_i) - \frac{w_1 v_i w_i}{v_1} - \frac{b_1 b_{i-1} v_i w_i}{v_1 w_{i-1}} = 0$$

$$\Rightarrow \frac{b_i}{1-\delta}(\frac{w_i b_{i-1}}{w_{i-1}} - b_i) - \frac{b_1 b_{i-1} v_i w_i}{v_1 w_{i-1}} = \frac{w_1 v_i w_i}{v_1} - \frac{v_i w_i v_{i-1}}{w_{i-1}} - v_i^2$$

$$\Rightarrow \frac{b_i}{1-\delta}(\frac{w_i b_{i-1}}{w_{i-1}} - b_i) - \frac{b_1 b_{i-1} v_i w_i}{v_1 w_{i-1}} = v_{i-1}^2 - v_i^2 + \frac{w_1 v_i w_i}{v_1}, \tag{12}$$

where Eq. (12) follows by substitution of Eq. (11). Rewriting the equation in case 3 using Eq. (10), we need to show that

$$v_{i-1}^2 - v_i^2 + \frac{w_1 v_i w_i}{v_1} + \frac{b_{i-1}}{1-\delta}(\frac{w_{i-1} b_i}{w_i} - b_{i-1}) - \frac{b_1 b_i v_{i-1} w_{i-1}}{v_1 w_i} \neq 0$$

and using Eq. (12), we can further simplify it to

$$\frac{b_i}{1-\delta}(\frac{w_i b_{i-1}}{w_{i-1}} - b_i) - \frac{b_1 b_{i-1} v_i w_i}{v_1 w_{i-1}} + \frac{b_{i-1}}{1-\delta}(\frac{w_{i-1} b_i}{w_i} - b_{i-1}) + \frac{b_1 b_i v_i w_i}{v_1 w_i} \neq 0$$

That expression can be rewritten as

$$\frac{b_1 v_i(b_i w_{i-1} - b_{i-1} w_i)}{v_1 w_{i-1}} + \frac{1}{1-\delta}(-b_{i-1}^2 + \frac{b_{i-1} b_i w_i}{w_{i-1}} + \frac{b_{i-1} b_i w_{i-1}}{w_i} - b_i^2)$$

The only way this expression is equal to 0 for every sufficiently small $\delta > 0$ is when both summands are 0. let us look at the second summand.

$$\frac{1}{1-\delta}(-b_{i-1}^2 + \frac{b_{i-1} b_i w_i}{w_{i-1}} + \frac{b_{i-1} b_i w_{i-1}}{w_i} - b_i^2) = \frac{1}{1-\delta}(-b_{i-1}^2 + b_{i-1} b_i(\frac{w_i}{w_{i-1}} + \frac{w_{i-1}}{w_i}) - b_i^2) \leq$$

$$\frac{1}{1-\delta}(-b_{i-1}^2 - 2b_{i-1} b_i - b_i^2) = -\frac{1}{1-\delta}(b_{i-1} + b_i)^2$$

Where the inequality stems from the inequality $x + \frac{1}{x} \leq -2$ for every $x < 0$ (and equality holds when $x = -1$) where in our case $x = \frac{w_i}{w_{i-1}}$, and they have different signs so $\frac{w_i}{w_{i-1}} < 0$. For the summand to be 0 it must holds that $w_i = -w_{i-1}$ and $b_i = -b_{i-1}$, but that can not happen because if that would have happened then $-\frac{b_i}{w_i} = -\frac{b_{i-1}}{w_{i-1}}$; i.e., the two neurons have the same breakpoint.

An example of a network with smaller norm can be found in Figure 2. $\qquad\square$

*Proof Of Thm. 3.4.* We prove that for each iteration, we add at least one training point to the set $S$. As the number of iteration is finite, and in each iteration the number of points added to $S$ are finite, $S$ is finite.

If the condition in line 6 in Algorithm 1 is met, by Thm. 3.2 one of the points added to $S$ must be a training point, and the number of such points is at most 4.

If both conditions at lines 9 and 11 are met, by Thm. 3.3 either $y$ or $z$ is a training point. So the ratio of training points in $S$ is at least $\frac{1}{4}$ $\qquad\square$

## B    PROOFS OF LEMMAS AND THEOREMS IN SECTION 4

We show an upper bound on the value $|\phi(\theta, \mathbf{x})|$ whenever $x$ is sampled according to $\mathcal{D}$ (that holds with high probability w.r.t the initialization and $\mathbf{x}$) and a lower bound whenever $\mathbf{x}$ is in the training set (that holds with high probability w.r.t the initialization). We prove that the lower bound is greater than the upper bound, thus giving us a way to differentiate between training and non training examples.

We use the same notations as the previous section.

NOTATIONS

- Let $J_+ = \{j : v_j > 0\}$ and $J_- = \{j : v_j < 0\}$.

- Let $m$ be the value of the network's margin.

- Let $\delta = \max_{i \neq j} \{|\mathbf{x}_i^\top \mathbf{x}_j|\}$ and $\Delta = \min_{i \in [n]} \{\|x_i\|^2\}$.

- For $\mathbf{x} \sim \mathcal{D}$ let $\delta_x = \max\{\delta, \max_i \in [n]\{|\mathbf{x}_i^\top \mathbf{x}|\}\}$.

The following 2 lemmas are taken from Frei et al. (2023b). In their paper, they proved a similar variant of the lemmas, and for the completeness of our proof, we give the proof of our variant.

**Lemma B.1.** *For all $l \in [n]$ we have*

$$max\left\{ \sum_{j \in J_+} v_j^2 \lambda_l \sigma'_{l,j}, \sum_{j \in J_-} v_j^2 \lambda_l \sigma'_{l,j} \right\} \leq \frac{m}{\Delta + 1 - 2\delta(n-1)}$$

*Proof.* Denote $\alpha_+ = \max_{i \in [n]} \left( \sum_{j \in J_+} v_j^2 \lambda_i \sigma'_{i,j} \right)$ and $\alpha_- = \max_{i \in [n]} \left( \sum_{j \in J_-} v_j^2 \lambda_i \sigma'_{i,j} \right)$. w.l.o.g $\alpha_+ \geq \alpha_-$ (other direction is similar). Denote $\alpha = \alpha_+$ and $k \in \text{argmax}_{i \in [n]} \left( \sum_{j \in J_+} v_j^2 \lambda_i \sigma'_{i,j} \right)$. If $\lambda_k = 0$ we are done. Otherwise, by KKT we know that $y_k \phi(\theta, x_k) = m$. By Eq. (8) and Eq. (9) we have for all $j$

$$\mathbf{w}_j^\top \mathbf{x}_k + b_j = \sum_{i=1}^n \lambda_i y_i \sigma'_{i,j} v_j (\mathbf{x}_i^\top \mathbf{x}_k + 1) = \lambda_k y_k \sigma'_{k,j} v_j (\|\mathbf{x}_k\|^2 + 1) + \sum_{i \neq k} \lambda_i y_i \sigma'_{i,j} v_j (\mathbf{x}_i^\top \mathbf{x}_k + 1)$$

$$\tag{13}$$

Consider 2 cases:

**CASE 1:** assume $y_k = 1$.

$$m = y_k \phi(\theta, \mathbf{x}_k) = \sum_{i=1}^{n} v_i \left[ \mathbf{w}_i^\top \mathbf{x}_k + b_i \right]_+$$

$$\geq \sum_{j \in J_+} v_j (\mathbf{w}_j^\top \mathbf{x}_k + b_j) + \sum_{j \in J_-} v_j \left[ \mathbf{w}_j^\top \mathbf{x}_k + b_j \right]_+ \tag{14}$$

Using the fact that $y_k = 1$ and Eq. (13) we get

$$\sum_{j \in J_+} v_j (\mathbf{w}_j^\top \mathbf{x}_k + b_j) = \sum_{j \in J_+} \left( \lambda_k \sigma'_{k,j} v_j^2 (\|\mathbf{x}_k\|^2 + 1) + \sum_{i \neq k} \lambda_i y_i \sigma'_{i,j} v_j^2 (\mathbf{x}_i^\top \mathbf{x}_k + 1) \right)$$

$$\geq \sum_{j \in J_+} \lambda_k v_j^2 \sigma'_{k,j} (\Delta + 1) - \delta \sum_{j \in J_+} \sum_{i \neq k} \lambda_i \sigma'_{i,j} v_j^2$$

$$\geq (\Delta + 1)\alpha - \delta(n-1)\alpha \tag{15}$$

Using $y_k = 1$ and Eq. (13) again we get

$$\sum_{j \in J_-} v_j \left[ \mathbf{w}_j^\top \mathbf{x}_k + b_j \right]_+ = \sum_{j \in J_-} v_j \left[ \lambda_k y_k \sigma'_{k,j} v_j (\|x_k\|^2 + 1) + \sum_{i \neq k} \lambda_i y_i \sigma'_{i,j} v_j (\mathbf{x}_i^\top \mathbf{x}_k + 1) \right]_+$$

$$\geq \sum_{j \in J_-} v_j \left[ \sum_{i \neq k} \lambda_i y_i \sigma'_{i,j} v_j (\mathbf{x}_i^\top \mathbf{x}_k + 1) \right]_+ = \sum_{j \in J_-} v_j \left[ \sum_{i \neq k} \lambda_i \sigma'_{i,j} |v_j| (\mathbf{x}_i^\top \mathbf{x}_k + 1) \right]_+$$

$$\geq \sum_{j \in j_-} v_j \left[ \sum_{i \neq k} \lambda_i \sigma'_{i,j} |v_j| (\delta + 1) \right]_+$$

$$\geq -(\delta + 1) \sum_{j \in j_-} \sum_{i \neq k} \lambda_i \sigma'_{i,j} v_j^2 \geq -(\delta + 1)(n-1)\alpha \tag{16}$$

Combining Eq. (14), Eq. (15) and Eq. (16) we get

$$m \geq (\Delta + 1)\alpha - \delta(n-1)\alpha - \delta(n-1)\alpha$$

$$= (\Delta + 1)\alpha - 2\delta(n-1)\alpha = \alpha(\Delta + 1 - 2\delta(n-1))$$

$$\Rightarrow \alpha \leq \frac{m}{\Delta + 1 - 2\delta(n-1)}$$

**CASE 2:** Assume $y_k = -1$.

Fix some $j \in J_+$. If $\sigma'_{j,k} = 0$ then

$$\lambda_k \sigma'_{k,j} v_j = 0 \leq \frac{\delta + 1}{\Delta + 1} \sum_{i \neq k} \lambda_i \sigma'_{i,j} v_j^2$$

Otherwise, by the definition of $\sigma'_{k,j}$ we have $\mathbf{w}_j^\top \mathbf{x}_k + b_j \geq 0$.

$$0 \leq \mathbf{w}_j^\top \mathbf{x}_k + b_j = \sum_{i \neq k} \lambda_i y_i \sigma'_{i,j} v_j (\mathbf{x}_i^\top \mathbf{x}_k + 1) + \lambda_k y_k \sigma'_{k,j} v_j (\|\mathbf{x}_k\|^2 + 1)$$

$$\leq \sum_{i \neq k} \lambda_i \sigma'_{i,j} v_j (\delta + 1) - \lambda_k \sigma'_{k,j} v_j (\Delta + 1)$$

$$\Rightarrow \lambda_k \sigma'_{k,j} v_j \leq \frac{\delta + 1}{\Delta + 1} \sum_{i \neq k} \lambda_i \sigma'_{i,j} v_j$$

$$\Rightarrow \lambda_k \sigma'_{k,j} v_j^2 \leq \frac{\delta + 1}{\Delta + 1} \sum_{i \neq k} \lambda_i \sigma'_{i,j} v_j^2$$

This is true for every $j \in J_+$, so by summing over all $j \in J_+$ we get

$$\sum_{j \in J_+} \lambda_k \sigma'_{k,j} v_j^2 \leq \frac{\delta+1}{\Delta+1} \sum_{j \in J_+} \sum_{i \neq k} \lambda_i \sigma'_{i,j} v_j^2$$

$$\leq \frac{\delta+1}{\Delta+1}(n-1) \cdot \max_{i \in [n]} \left( \sum_{j \in J_+} \lambda_i \sigma'_{i,j} v_j^2 \right)$$

$$< \max_{i \in [n]} \left( \sum_{j \in J_+} \lambda_i \sigma'_{i,j} v_j^2 \right) = \sum_{j \in J_+} \lambda_k \sigma'_{k,j} v_j^2$$

Where the last equality is the definition of $k$. This case can not happen, so $y_k = 1$ and we have already proved that case. □

**Lemma B.2.** *For all $l \in [n]$ such that $y_l = 1$ we have*

$$\sum_{j \in J_+} v_j^2 \lambda_l \sigma'_{l,j} \geq \left( m - (\delta+1)(n-1)\frac{m}{\Delta+1-2\delta(n-1)} \right) \cdot \frac{1}{\Delta+1}$$

*and for all $l \in [n]$ such that $y_l = -1$ we have*

$$\sum_{j \in J_-} v_j^2 \lambda_l \sigma'_{l,j} \geq \left( m - (\delta+1)(n-1)\frac{m}{\Delta+1-2\delta(n-1)} \right) \cdot \frac{1}{\Delta+1}$$

*Proof.* Let $k \in [n]$ such that $y_k = 1$. We have

$$m \leq \phi(\theta, x_k) = \sum_{j \in J} v_j \left[ \mathbf{w}_j^\top \mathbf{x}_k + b_j \right]_+ \leq \sum_{j \in J+} v_j \left[ \mathbf{w}_j^\top \mathbf{x}_k + b_j \right]_+ \leq \sum_{j \in J_+} v_j |\mathbf{w}_j^\top \mathbf{x}_k + b_j|$$

Let us upper bound it

$$\sum_{j \in J_+} v_j \left| \lambda_k y_k \sigma'_{k,j} v_j (\|\mathbf{x}_k\|^2 + 1) + \sum_{i \neq k} \lambda_i y_i \sigma'_{i,j} v_j (\mathbf{x}_i^\top \mathbf{x}_k + 1) \right|$$

$$\leq \sum_{j \in J_+} v_j \left( \lambda_k \sigma'_{k,j} v_j (\|\mathbf{x}_k\|^2 + 1) + \sum_{i \neq k} \lambda_i \sigma'_{i,j} v_j |\mathbf{x}_i^\top \mathbf{x}_k + 1| \right)$$

$$= \sum_{j \in J_+} \left( \lambda_k \sigma'_{k,j} v_j^2 (\|\mathbf{x}_k\|^2 + 1) + \sum_{i \neq k} \lambda_i \sigma'_{i,j} v_j^2 |\mathbf{x}_i^\top \mathbf{x}_k + 1| \right)$$

$$\leq \sum_{j \in J_+} \left( (\Delta+1)\lambda_k \sigma'_{k,j} v_j^2 + (\delta+1) \sum_{i \neq k} \lambda_i \sigma'_{i,j} v_j^2 \right)$$

$$= (\Delta+1) \sum_{j \in J_+} \lambda_k \sigma'_{k,j} v_j^2 + (\delta+1) \sum_{j \in J_+} \sum_{i \neq k} \lambda_i \sigma'_{i,j} v_j^2$$

Using B.1 we get

$$m \leq (\Delta+1) \sum_{j \in J_+} \lambda_k \sigma'_{k,j} v_j^2 + (\delta+1)(n-1)\frac{m}{\Delta+1-2\delta(n-1)}$$

$$\Rightarrow \sum_{j \in J_+} \lambda_k \sigma'_{k,j} v_j^2 \geq \left( m - (\delta+1)(n-1)\frac{m}{\Delta+1-2\delta(n-1)} \right) \cdot \frac{1}{\Delta+1}$$

Similar arguments yield the other inequality □

**Lemma B.3.** *Under Assumption 4.1, with probability at least $1 - 2\tau$*

$$O(\frac{n \cdot \delta}{\Delta}) = o_d(1)$$

*Proof.* First, we prove using the union bound that $\Pr[n \cdot \delta \geq \Omega(d)] < \tau$.

$$\Pr[n \cdot \delta \geq \Omega(d)] \leq \sum_{\substack{i,j=1 \\ i \neq j}}^{n} \Pr[n \cdot |\mathbf{x}_i^\top \mathbf{x}_j| \geq \Omega(d)] \leq \binom{n}{2} \cdot \frac{\tau}{n^2} < \tau$$

Secondly, we prove using the union bound that $\Pr[\Delta < o(d)] < \tau$

$$\Pr[\Delta < o(d)] \leq \sum_{i=1}^{n} \Pr[\|\mathbf{x}_i\|^2 < o(d)] \leq n \cdot \frac{\tau}{n} = \tau$$

Now, using the union bound again, we get

$$\Pr[\frac{n \cdot \delta}{\Delta} > \Omega_d(1)] \leq \Pr[\Delta < o(d)] + \Pr[n \cdot \delta \geq \Omega(d)] \leq 2\tau$$

And hence with probability at least $1 - 2\tau$ we have that

$$O(\frac{n \cdot \delta}{\Delta}) = o_d(1)$$

$\square$

**Lemma B.4.** *Let $x \sim \mathcal{D}$. Under Assumption 4.1, with probability at least $1 - 2\tau$*

$$O(\frac{n \cdot \delta_x}{\Delta}) = o_d(1)$$

*Proof.* First, we prove using the union bound that $\Pr[\Delta < o(d)] < \tau$

$$\Pr[\Delta < o(d)] \leq \sum_{i=1}^{n} \Pr[\|\mathbf{x}_i\|^2 < o(d)] \leq n \cdot \frac{\tau}{n} = \tau$$

Second, we prove using the union bound that $\Pr[n \cdot \delta_x \geq \Omega(d)] < \tau$.

$$\Pr[n \cdot \delta_x \geq \Omega(d)] \leq \sum_{\substack{i,j=1 \\ i \neq j}}^{n} \Pr[n \cdot |\mathbf{x}_i^\top \mathbf{x}_j| \geq \Omega(d)] + \sum_{i=1}^{n} \Pr[n|\mathbf{x}_i^\top \mathbf{x}| \geq \Omega(d)]$$

$$\leq \binom{n+1}{2} \cdot \frac{\tau}{n^2} < \tau$$

Where in last inequality we used the fact that $n \geq 3$. Now, using the union bound again, we get

$$\Pr[\frac{n \cdot \delta_x}{\Delta} > \Omega_d(1)] \leq \Pr[\Delta < o(d)] + \Pr[n \cdot \delta_x \geq \Omega(d)] \leq 2\tau$$

And hence with probability at least $1 - 2\tau$ we have that

$$O(\frac{n \cdot \delta_x}{\Delta}) = o_d(1)$$

$\square$

*Proof Of Thm. 4.2.* Assume $\mathbf{x}$ is in the training data, i.e there is $k \in [n]$ such that $\mathbf{x} = \mathbf{x}_k$. Assume w.l.o.g that $\phi(\theta, \mathbf{x}_k) > 0$, i.e $y_k = 1$ (the case $y_k = -1$ is similar).
We can decompose the network into 2 components, as follow:

$$\phi(\theta, \mathbf{x}_k) = \sum_{i=1}^{n} v_i \left[\mathbf{w}_i^\top \mathbf{x}_k + b_i\right]_+ = \sum_{j \in J_+} v_j \left[\mathbf{w}_j^\top \mathbf{x}_k + b_j\right]_+ + \sum_{j \in J_-} v_j \left[\mathbf{w}_j^\top \mathbf{x}_k + b_j\right]_+ \quad (17)$$

Let us bound every sum from below. Using Eq. (8), Eq. (9) and the fact that $[x]_+ \geq x$ we get

$$\sum_{j \in J_+} v_j \left[ \mathbf{w}_j^\top \mathbf{x}_k + b_j \right]_+ \geq \sum_{j \in J_+} v_j (\mathbf{w}_j^\top \mathbf{x}_k + b_j) = \sum_{j \in J_+} v_j \left[ \sum_{i=1}^n \lambda_i y_i v_j \sigma'_{i,j} (\mathbf{x}_i^\top \mathbf{x}_k + 1) \right]$$

$$= \sum_{j \in J_+} v_j^2 \lambda_k y_k \sigma'_{k,j} (\|\mathbf{x}_k\|^2 + 1) + \sum_{j \in J_+} \sum_{i \neq k} v_j^2 \lambda_i y_i \sigma'_{i,j} (\mathbf{x}_i^\top \mathbf{x}_k + 1)$$

$$\geq \sum_{j \in J_+} v_j^2 \lambda_k \sigma'_{k,j} (\|\mathbf{x}_k\|^2 + 1) - \sum_{j \in J_+} \sum_{i \neq k} v_j^2 \lambda_i \sigma'_{i,j} |\mathbf{x}_i^\top \mathbf{x}_k + 1| \tag{18}$$

And for the second sum

$$\sum_{j \in J_-} v_j \left[ \mathbf{w}_j^\top \mathbf{x}_k + b_j \right]_+ = \sum_{j \in J_-} v_j \left[ \sum_{i=1}^n \lambda_i y_i v_j \sigma'_{i,j} (\mathbf{x}_i^\top \mathbf{x}_k + 1) \right]_+$$

$$= \sum_{j \in J_-} v_j \left[ \lambda_k y_k v_j \sigma'_{k,j} (\|\mathbf{x}_k\|^2 + 1) + \sum_{i \neq k} \lambda_i y_i v_j \sigma'_{i,j} (\mathbf{x}_i^\top \mathbf{x}_k + 1) \right]_+$$

$$\geq \sum_{j \in J_-} v_j \left[ \sum_{i \neq k} \lambda_i y_i v_j \sigma'_{i,j} (\mathbf{x}_i^\top \mathbf{x}_k + 1) \right]_+ \tag{19}$$

We need to show that $\sum_{j \in J_+} v_j^2 \lambda_k \sigma'_{k,j}$, $\sum_{j \in J_+} \sum_{i \neq k} v_j^2 \lambda_i \sigma'_{i,j}$, $\sum_{j \in J_-} \sum_{i \neq k} \lambda_i y_i v_j^2 \sigma'_{i,j}$ and $\sum_{j \in J_-} v_j^2 \lambda_k \sigma'_{k,j}$ are not too small and not too large.

From Lemma B.2 we have that

$$\sum_{j \in J_+} v_j^2 \lambda_k \sigma'_{k,j} \geq \left( m - (\delta + 1)(n-1) \frac{m}{\Delta + 1 - 2\delta(n-1)} \right) \cdot \frac{1}{\Delta + 1}$$

By Lemma B.3 we have that with probability at least $1 - 2\tau$

$$O(\frac{n \cdot \delta}{\Delta}) = O(\frac{n \cdot \delta}{\Delta}) = o_d(1)$$

which means that

$$\left( m - (\delta + 1)(n-1) \frac{m}{\Delta + 1 - 2\delta(n-1)} \right) \cdot \frac{1}{\Delta + 1} > 0$$

which means that $\lambda_k > 0$, which means that $x_k$ is on the margin and hence $\phi(\theta, x_k) = m$.
If $\mathbf{x}$ is not a training point, then

$$|\phi(\theta, x)| = \left| \sum_{j \in J_+} v_j \sum_{i \in [n]} \lambda_i y_i \sigma'_{i,j} v_j (\mathbf{x}_i^\top \mathbf{x} + 1) + \sum_{j \in J_-} v_j \sum_{i \in [n]} \lambda_i y_i \sigma'_{i,j} v_j (\mathbf{x}_i^\top \mathbf{x} + 1) \right|$$

$$\leq \sum_{j \in J_+} \sum_{i \in [n]} \lambda_i \sigma'_{i,j} v_j^2 |\mathbf{x}_i^\top \mathbf{x} + 1| + \sum_{j \in J_-} \sum_{i \in [n]} \lambda_i \sigma'_{i,j} v_j^2 |\mathbf{x}_i^\top \mathbf{x} + 1|$$

$$\leq 2 \cdot n \cdot (\delta_x + 1) \cdot \frac{m}{\Delta + 1 - 2\delta \cdot (n-1)} \leq 2 \cdot n \cdot (\delta_x + 1) \cdot \frac{m}{\Delta + 1 - 2\delta_x \cdot (n-1)} = O(\frac{n \cdot m \cdot \delta_x}{\Delta})$$

Where in the second inequality we used Lemma B.1. By Lemma B.4 we have that with probability at least $1 - 2\tau$

$$2 \cdot n \cdot (\delta_x + 1) \cdot \frac{m}{\Delta + 1 - 2\delta_x \cdot (n-1)} = O(\frac{n \cdot m \cdot \delta_x}{\Delta}) = m \cdot O(\frac{n \cdot \delta_x}{\Delta}) = o_d(m)$$

$\square$

**Remark B.5** (On the lower bound of the margin). *From Thm. 4.2 we know that w.h.p. at least $\frac{n}{2}$ training points lie on the margin. Our loss function is*

$$\ell(\Phi(\theta; \mathbf{x}) \cdot y) = \log(1 + e^{-y \cdot \Phi(\theta; \mathbf{x})})$$

*so we have that*

$$\frac{1}{2e} > L(\boldsymbol{\theta}) = \frac{1}{n} \sum_{i=1}^{n} \ell(\Phi(\mathbf{x}_i), y_i) \geq \frac{1}{n} \cdot \frac{n}{2} \cdot \log(1 + e^{-m})$$

$$\Rightarrow \frac{1}{e} > \log(1 + e^{-m})$$

*Now we can extract a lower bound on $m$:*

$$\log(1 + e^{-m}) < \frac{1}{e} \Rightarrow 1 + e^{-m} < e^{e^{-1}} \Rightarrow e^{-m} < e^{e^{-1}} \Rightarrow m > \frac{1}{e}.$$

*Same argument shows a similar bound for the exponential loss $\ell(x) = e^{-x}$.*

**Remark B.6** (On the upper bound of the margin). *When training a neural network using gradient-based methods, the training process usually halts once the gradient is sufficiently small. When considering the exponential or logistic losses as in our case, a large margin implies small loss which in turn implies that the gradient is small. This suggests that making further progress when the margin is large becomes very difficult, and the training process is expected to halt. More formally, recall the logistic loss function (a similar argument implies the same result for the exponential loss):*

$$\ell(\Phi(\boldsymbol{\theta}; \mathbf{x}) \cdot y) = \log(1 + e^{-y \cdot \Phi(\boldsymbol{\theta}; \mathbf{x})}).$$

*This function is monotonically decreasing in the expression $y\Phi(\theta; \mathbf{x})$, so the loss is maximized for points that are on the margin, and we can upper bound*

$$\left| \frac{\partial \ell(\Phi(\boldsymbol{\theta}; \mathbf{x}) \cdot y)}{\partial \Phi(\boldsymbol{\theta}; \mathbf{x})} \right| = \left| \frac{-y \cdot \Phi(\boldsymbol{\theta}; \mathbf{x}) \cdot e^{-y \cdot \Phi(\boldsymbol{\theta}; \mathbf{x})}}{1 + e^{-y \cdot \Phi(\boldsymbol{\theta}; \mathbf{x})}} \right| \leq \left| \frac{m e^{-m}}{1 + e^{-m}} \right|.$$

*The above yields*

$$\left| \frac{\partial L(\boldsymbol{\theta})}{\partial \boldsymbol{\theta}_j} \right| \leq \frac{1}{n} \sum_{i=1}^{n} \left| \frac{\partial \ell(\Phi(\boldsymbol{\theta}; \mathbf{x}_i) \cdot y_i)}{\partial \Phi(\boldsymbol{\theta}; \mathbf{x}_i)} \right| \cdot \left| \frac{\partial \Phi(\boldsymbol{\theta}; \mathbf{x}_i)}{\partial \boldsymbol{\theta}_j} \right| \leq \text{poly}(d) \cdot \left| \frac{m e^{-m}}{1 + e^{-m}} \right|,$$

*which allows us to bound the norm of the gradient by:*

$$\|\nabla_{\boldsymbol{\theta}} L(\boldsymbol{\theta})\| \leq w \cdot \text{poly}(d) \cdot \left| \frac{m e^{-m}}{1 + e^{-m}} \right| = \text{poly}(d) \cdot \left| \frac{m e^{-m}}{1 + e^{-m}} \right|,$$

*where $w$ denotes the width of the network which we assume to be polynomial in $d$ (since otherwise even making a prediction is computationally inefficient).*

*If, for example, the margin is $m = \log^2 d = o(\sqrt{d})$, we get that*

$$\|\nabla_{\boldsymbol{\theta}} L(\boldsymbol{\theta})\| \leq \text{poly}(d) \left| \frac{\log^2 d e^{-\log^2 d}}{1 + e^{-\log^2 d}} \right| \leq \text{poly}(d) \log^2 d \cdot e^{-\log^2 d} = \text{poly}(d) \log^2 d \cdot d^{-\log d},$$

*which is smaller than any inverse polynomial in $d$. Hence, if we train for at most polynomially many iterations and label all the data points correctly (i.e. the margin is strictly positive), then training effectively stops when the margin reaches $O(\log^2 d) = o(\sqrt{d})$, and all the data points on the margin (which consist of at least one point) will have an output of magnitude $O(polylog(d))$.*

## C  HIGH-DIMENSIONAL ATTACKS IN THE STATISTICALLY LEARNABLE CASE

In this appendix, we show that Item 3 exemplifies a setting where Assumption 4.1 is satisfied, yet the distribution being considered is statistically learnable. This was shown in several recent works, which considered the optimization of a shallow neural network, in a setting similar to ours.

Consider for example the setting studied in Xu et al. (2023). In that paper, the authors prove a generalization result under the assumption of a certain target distribution of a mixture of four Gaussians. Such a distribution is captured by Item 3 in our examples for distributions which satisfy Assumption 4.1, which indicates that our proposed membership inference attack will work. Specifically, to make sure that both Assumption 4.1 and the requirements made in Xu et al. (2023) are satisfied, it must hold in addition that:

- The norm of each mean satisfies $\|\boldsymbol{\mu}^{(i)}\|^2 \geq \Omega(n^{0.51}\sqrt{d})$.

- The dimension of the feature space satisfies $d \geq \Omega(n^2 \max\{\|\boldsymbol{\mu}^{(i)}\|^2\})$.

- The number of neurons satisfies $k \geq \Omega(n^{0.02})$.

A bit more precisely, their theorem states the following:

**Theorem C.1** (Xu et al. (2023), Theorem 3.1, informal)**.** *Suppose that the above assumptions are satisfied, then with high probability over the training set and the initialization of the weights, we have*

$$\Pr_{(\mathbf{x},y)\sim\mathcal{D}}[y \neq \text{sign}(\phi(\boldsymbol{\theta},\mathbf{x}))] \leq \exp(-\Omega(n^{2.01}))$$

These assumptions essentially imply Assumption 4.1.

Similarly, Assumption 4.1, and specifically Item 3 in our examples, also holds in other settings where generalization was proved in previous works:

- Xu and Gu (2023); Frei et al. (2022); Chatterji and Long (2021) proved generalization in a setting where the data distribution consists of two opposite Gaussians (or more broadly in an even more general setting) with covariance $I_d$ and means $\pm\boldsymbol{\mu}$, where $\|\boldsymbol{\mu}\| = d^\beta$ with $\beta \in (0.25, 0.5)$. Their sample size is $n = \tilde{\Omega}(1)$. This setting satisfies our condition from Item 3. Specifically, the result of Xu and Gu (2023) holds for 2-layer ReLU networks.

- In Frei et al. (2023a) (see the discussion after Theorem 11 therein), the authors mention two specific settings that satisfy their theorem requirements, and thus good generalization performance can be achieved (and more specifically, in Corollaries 12 and 13, they further show that in these settings good generalization is achieved by the max-margin linear predictor and by a trained 2-layer leaky-ReLU network). Note that these settings satisfy our condition from Item 3.

# D PROOFS OF DISTRIBUTIONS

In this section we prove the examples in section 4.

**Uniform Distribution** For the uniform distribution on $\sqrt{d} \cdot \mathbb{S}^{d-1}$, the next lemma shows why is satisfies our assumptions.
The lemma is from Vardi et al. (2022a), and we give a paraphrased version of it for the sake of the reader.

**Lemma D.1.** *Let* $\mathbf{x}, \mathbf{y} \sim U(\sqrt{d} \cdot \mathbb{S}^{d-1})$. *Then, with probability at least* $1 - d^{1-\ln(d)/4} = 1 - o_d(1)$ *we have* $|\langle \mathbf{x}, \mathbf{y} \rangle| \leq \sqrt{d} \cdot \log d = o(d)$.

**Remark D.2.** *For the uniform distribution, the training set size can be* $n = o\left(\frac{\sqrt{d}}{\log d}\right)$ *and* $\tau = n^2 \cdot d^{1-\ln(d)/4} = o_d(1)$

**Normal Distribution** As for the normal distribution, the following two lemmas prove its correctness

**Lemma D.3.** *Let* $\mathcal{N} = \mathcal{N}(\boldsymbol{\mu}, I)$ *be a normal distribution on* $\mathbb{R}^d$. *Let* $\mathbf{x}, \mathbf{y} \sim \mathcal{N}(\boldsymbol{\mu}, I)$. *Assume that* $\|\boldsymbol{\mu}\|^2 = o(d)$. *then with probability at least*

$$1 - 2\exp\left(-\frac{c_1}{16c_2^2} \cdot \frac{d^{2\epsilon}}{\|\mu\|^2}\right) - \max\left(2\exp\left(-\frac{c_1}{2c_2^2}d^\epsilon\right), 2\exp\left(-\frac{c_1}{4c_2^4} \cdot d^{2\epsilon-1}\right)\right)$$

$$- \max\left(2\exp\left(-\frac{c_1}{c_2^4}d^{2\epsilon-1}\right), 2\exp\left(-\frac{c_1}{c_2^2}d^\epsilon\right)\right)$$

$$= 1 - o_d(1)$$

*we have* $|\langle \mathbf{x}, \mathbf{y} \rangle| = o(d)$ *and* $\|\mathbf{x}\|^2 = O(d)$, *where* $c_1$, $c_2$ *are constants independent of $d$, and* $\frac{1}{2} < \epsilon < 1$.

*Proof.* Let $\mathbf{x}, \mathbf{y} \sim \mathcal{N}(\boldsymbol{\mu}, \Sigma)$ independently.

W.l.o.g $\Sigma$ is diagonal, otherwise there is a unitary matrix $U$ such that $U\mathbf{x}, U\mathbf{y} \sim \mathcal{N}(U\boldsymbol{\mu}, U\Sigma U^\top)$ where $U\Sigma U^\top$ is diagonal. Since $U$ is unitary we have that

$$\langle U\mathbf{x}, U\mathbf{y} \rangle = \langle \mathbf{x}, \mathbf{y} \rangle$$
$$\|U\mathbf{x}\| = \|\mathbf{x}\|$$

So we can assume that $\Sigma$ is diagonal.

For comfort, we define some notations:

- The sub-Gaussian norm $\| \cdot \|_{\psi_2}$ for a sub-Gaussian random variable $\mathbf{x}$ is defined by

$$\|\mathbf{x}\|_{\psi_2} = \inf \left\{ t > 0 : E\left[\exp\left(\frac{\mathbf{x}^2}{t}\right)\right] \le 2 \right\}$$

- The sub-exponential norm $\| \cdot \|_{\psi_1}$ for a sub-exponential random variable $\mathbf{x}$ is defined by

$$\|\mathbf{x}\|_{\psi_1} = \inf \left\{ t > 0 : E\left[\exp\left(\frac{|\mathbf{x}|}{t}\right)\right] \le 2 \right\}$$

First, let us compute $E\left[\|\mathbf{x}\|^2\right]$. Note that

$$\|\mathbf{x}\|^2 = \sum_{i=1}^{d} \mathbf{x}_i^2,$$

then $E[\mathbf{x}_i^2] = E[\mathbf{x}_i]^2 + \mathrm{Var}(\mathbf{x}_i) = \boldsymbol{\mu}_i^2 + 1$

$$E\left[\|\mathbf{x}\|^2\right] = E\left[\sum_{i=1}^{d} \mathbf{x}_i^2\right] = \sum_{i=1}^{d} E[\mathbf{x}_i^2] = \sum_{i=1}^{d} \mathrm{Var}(\mathbf{x}_i) + \boldsymbol{\mu}_i^2 = \mathrm{tr}(I) + \|\boldsymbol{\mu}\|^2 = O(d)$$

Note that we can write $\mathbf{x}$ as $\mathbf{x} = \boldsymbol{\mu} + \mathbf{z}$ where $\mathbf{z} \sim \mathcal{N}(0, I)$. We can write $\|\mathbf{x}\|^2 = \|\boldsymbol{\mu} + \mathbf{z}\|^2 = \|\boldsymbol{\mu}\|^2 + 2|\boldsymbol{\mu}^\top \mathbf{z}| + \|\mathbf{z}\|^2$. So we need to upper bound

$$\|\mathbf{x}\|^2 - E\left[\|\mathbf{x}\|^2\right] = \|\boldsymbol{\mu}\|^2 + 2\boldsymbol{\mu}^\top \mathbf{z} + \|\mathbf{z}\|^2 - \|\boldsymbol{\mu}\|^2 - 2\boldsymbol{\mu}^\top E[\mathbf{z}] - E\left[\|z\|^2\right] = \|\mathbf{z}\|^2 - E\left[\|z\|^2\right] + 2\boldsymbol{\mu}^\top \mathbf{z}$$

Where in the last equality we used the fact that $E[\mathbf{z}] = 0$

From the union bound we get that for every $t > 0$

$$\Pr\left[\left|\mathbf{x}^2 - E[\|\mathbf{x}\|^2]\right| > t\right] = \Pr\left[\left|\|\mathbf{z}\|^2 - E[\|\mathbf{z}\|^2] + 2\boldsymbol{\mu}^\top \mathbf{z}\right| > t\right]$$
$$\le \Pr\left[\left|\|\mathbf{z}\|^2 - E[\|\mathbf{z}\|^2]\right| + 2\left|\boldsymbol{\mu}^\top \mathbf{z}\right| > t\right]$$
$$\le \Pr\left[\left|\|\mathbf{z}\|^2 - E[\|\mathbf{z}\|^2]\right| > \frac{t}{2}\right] + \Pr\left[2\left|\boldsymbol{\mu}^\top \mathbf{z}\right| > \frac{t}{2}\right]$$

Let us bound the first term. To do so, we use Hanson-Wright Inequality (Vershynin (2018) Theorem 6.2.1).

$$\Pr\left[\left|\|\mathbf{z}\|^2 - E[\|\mathbf{z}\|^2]\right| > \frac{t}{2}\right] \le 2\exp\left[-c_1 \min\left(\frac{t^2}{4 \cdot K^4 \cdot d}, \frac{t}{2 \cdot K^2}\right)\right]$$

Where $K = \max_i \|\mathbf{x}_i\|_{\psi_2} = c_2$ and $c_1, c_2$ are constant independent of $d$. We set $t = d^\epsilon$ for $\frac{1}{2} < \epsilon < 1$.

**Case 1 -** $\frac{t^2}{4 \cdot K^4 \cdot d}$ **is the minimum**

$$\Pr\left[\left|\|\mathbf{z}\|^2 - E[\|\mathbf{z}\|^2]\right| > \frac{t}{2}\right] \le 2\exp\left(-c_1 \frac{d^{2\epsilon}}{c_2^4 \cdot 4 \cdot d}\right) = 2\exp\left(-\frac{c_1}{4 \cdot c_2^4} \cdot d^{2\epsilon - 1}\right) = o_d(1)$$

**Case 2 - $\frac{t}{2 \cdot K^2}$ is the minimum**

$$\Pr\left[\left|\|\mathbf{z}\|^2 - E[\|\mathbf{z}\|^2]\right| > \frac{t}{2}\right] \leq 2\exp\left(-c_1\frac{d^\epsilon}{2 \cdot c_2^2}\right) = o_d(1)$$

Now we upper bound the term $\Pr\left[2|\boldsymbol{\mu}^\top\mathbf{z}| > \frac{t}{2}\right] = \Pr\left[|\boldsymbol{\mu}^\top\mathbf{z}| > \frac{t}{4}\right]$.
From General Hoeffding's inequality (Vershynin (2018) Theorem 2.6.3) we get that

$$\Pr\left[|\boldsymbol{\mu}^\top\mathbf{z}| > \frac{t}{4}\right] \leq 2\exp\left(-\frac{c_1 t^2}{16 \cdot K^2 \cdot \|\boldsymbol{\mu}\|^2}\right)$$

Where $K = \max_i \|\mathbf{x}_i\|_{\psi_2} = c_2$ and $c_1$, $c_2$ are constant independent of $d$. Putting it all together we get

$$\Pr\left[|\boldsymbol{\mu}^\top\mathbf{z}| > \frac{t}{4}\right] \leq 2\exp\left(-\frac{c_1 t^2}{16 \cdot K^2 \cdot \|\boldsymbol{\mu}\|^2}\right)$$

$$= 2\exp\left(-\frac{c_1}{16c_2^2}\frac{d^{2\epsilon}}{\|\boldsymbol{\mu}\|^2}\right) = 2\exp\left(-\frac{c_1}{16c_2^2}\frac{d^{2\epsilon}}{\|\boldsymbol{\mu}\|^2}\right) = o_d(1)$$

Where in last inequality we used the fact that $2\epsilon > 1$.

All in all, we showed that $E[\|\mathbf{x}\|^2] = O(d)$ and that with probability

$$1 - \max\left(2\exp\left(-\frac{c_1}{4c_2^2} \cdot d^{2\epsilon-1}\right), 2\exp\left(-\frac{c_1}{2c_2^2} \cdot d^\epsilon\right)\right) - 2\exp\left(-\frac{c_1}{16c_2^2}\frac{d^{2\epsilon}}{\|\mu\|^2}\right) = 1 - o_d(1)$$

we have that

$$\left|\|\mathbf{x}\|^2 - E[\|\mathbf{x}\|^2]\right| < d^\epsilon = o(d)$$

and specifically $\|\mathbf{x}\|^2 = O(d)$

Since $\mathbf{x}$ is normal, each $\mathbf{x}_i$ is sub-Gaussian (and the same for $\mathbf{y}$).
Let us have a look at $\mathbf{x}^\top\mathbf{y}$: Since $\mathbf{x}_i, \mathbf{y}_i$ are sub-Gaussians, $\mathbf{x}_i \cdot \mathbf{y}_i$ is sub-exponential (Vershynin (2018), Lemma 2.7.7). It is also known that a sum of sub-exponential random variables is in itself sub-exponential, so we get that

$$\mathbf{x}^\top\mathbf{y} = \sum_{i=1}^d x_i y_i$$

is sub-exponential. By the centering lemma (Vershynin (2018) Exercise 2.7.10), $x_i y_i - E[x_i y_i] = x_i y_i - \mu_i^2$ is also sub-exponential, with mean zero. We can use Bernstein's inequality (Vershynin (2018), Theorem 2.8.1) to get:

$$\Pr\left[|\mathbf{x}^\top\mathbf{y} - \|\boldsymbol{\mu}\|^2| > t\right] = \Pr\left[\left|\sum_{i=1}^d x_i y_i - \mu_i^2\right| > t\right]$$

$$\leq 2\exp\left[-c_1 \cdot \min\left(\frac{t}{\max_i \|x_i y_i - \mu_i\|_{\psi_1}}, \frac{t^2}{\sum_{i=1}^d \|x_i y_i - \mu_i\|_{\psi_1}^2}\right)\right]$$

$$\leq 2\exp\left[-c_1 \cdot \min\left(\frac{t}{\max_i \|x_i y_i\|_{\psi_1}}, \frac{t^2}{\sum_{i=1}^d \|x_i y_i\|_{\psi_1}^2}\right)\right]$$

$$\leq 2\exp\left[-c_1 \cdot \min\left(\frac{t}{\max_i \|x_i\|_{\psi_2}\|y_i\|_{\psi_2}}, \frac{t^2}{\sum_{i=1}^d \|x_i\|_{\psi_2}^2\|y_i\|_{\psi_2}^2}\right)\right]$$

$$= 2\exp\left[-c_1 \cdot \min\left(\frac{t}{c_2^2}, \frac{t^2}{\sum_{i=1}^d c_2^4}\right)\right]$$

Where $c_1$, $c_2$ are constants that do not depend on the dimension $d$. In the second inequality we used the fact that $\|\mathbf{x} - E[\mathbf{x}]\|_{\psi_1} \leq \|\mathbf{x}\|_{\psi_1}$ (Vershynin (2018) Exercise 2.7.10) and in the third inequality we used the fact that $\|x_i y_i\|_{\psi_1} \leq \|x_i\|_{\psi_2}\|y_i\|_{\psi_2}$ (Vershynin (2018) Lemma 2.7.7). Setting $t = d^\epsilon$ for some $\frac{1}{2} < \epsilon < 1$ we get:

**Case 1 -** $\frac{t}{c_2^2}$ **is the minimum**

$$\Pr[|\mathbf{x}^\top \mathbf{y} - \|\boldsymbol{\mu}\|^2| > d^\epsilon] \leq 2\exp\left[-c_1 \cdot \frac{d^\epsilon}{c_2^2}\right] = o_d(1)$$

And since both $\|\boldsymbol{\mu}\|^2 = o(d)$ and $d^\epsilon = o(d)$ we get that w.h.p. $\mathbf{x}^\top \mathbf{y} = o(d)$

**Case 2 -** $\frac{t^2}{\sum_{i=1}^d c_2^4}$ **is the minimum**

$$\Pr[|\mathbf{x}^\top \mathbf{y} - \|\boldsymbol{\mu}\|^2| > d^\epsilon] \leq 2\exp\left[-c_1 \cdot \frac{d^{2\epsilon}}{c_2^4 \cdot d}\right]$$

$$= 2\exp\left[-\frac{c_1}{c_2^4} \cdot d^{2\epsilon - 1}\right] = o_d(1)$$

Using the union bound, with probability at least

$$1 - 2\exp\left(-\frac{c_1}{16c_2^2} \cdot \frac{d^{2\epsilon}}{\|\mu\|^2}\right) - \max\left(2\exp\left(-\frac{c_1}{2c_2^2}d^\epsilon\right), 2\exp\left(-\frac{c_1}{4c_2^4} \cdot d^{2\epsilon - 1}\right)\right)$$

$$- \max\left(2\exp\left(-\frac{c_1}{c_2^4}d^{2\epsilon - 1}\right), 2\exp\left(-\frac{c_1}{c_2^2}d^\epsilon\right)\right)$$

$$= 1 - o_d(1)$$

we have $|\langle \mathbf{x}, \mathbf{y} \rangle| = o(d)$ and $\|\mathbf{x}\|^2 = O(d)$. $\qquad \square$

**Remark D.4.** *we want $n \cdot |\mathbf{x}^\top \mathbf{y}| = o(d)$ to hold, so*

$$n \cdot |\mathbf{x}^\top \mathbf{y}| \leq n \cdot (\|\mu\|^2 + d^\epsilon) = o(d) \Rightarrow n = \frac{o(d)}{\|\mu\|^2 + d^\epsilon}$$

**Lemma D.5.** *Let $\mathcal{N} = \mathcal{N}(\boldsymbol{\mu}, I)$ be a normal distribution on $\mathbb{R}^d$. Let $\mathbf{x}, \mathbf{y} \sim \mathcal{N}(\boldsymbol{\mu}, I)$. Assume that $\|\boldsymbol{\mu}\|^2 = o(d)$, and $n = \frac{o(d)}{\|\mu\|^2 + d^\epsilon}$ for $\frac{1}{2} < \epsilon < 1$. Denote*

$$k = 2\exp\left(-\frac{c_1}{16c_2^2} \cdot \frac{d^{2\epsilon}}{\|\mu\|^2}\right) + \max\left(2\exp\left(-\frac{c_1}{2c_2^2}d^\epsilon\right), 2\exp\left(-\frac{c_1}{4c_2^4} \cdot d^{2\epsilon - 1}\right)\right)$$

$$+ \max\left(2\exp\left(-\frac{c_1}{c_2^4}d^{2\epsilon - 1}\right), 2\exp\left(-\frac{c_1}{c_2^2}d^\epsilon\right)\right)$$

*where $c_1$, $c_2$ are the constants from Lemma D.3. Let $\tau = k \cdot n$. Then with probability at least $1 - \frac{\tau}{n^2}$ have $|n \cdot \langle \mathbf{x}, \mathbf{y} \rangle| = o(d)$ and $\|\mathbf{x}\|^2 = O(d)$. In particular, those $n$ and $\tau$ satisfy Assumption 4.1.*

*Proof.* From Lemma D.3 we know that with probability at least $1 - k$ we have that $|\langle \mathbf{x}, \mathbf{y} \rangle| \leq \|\mu\|^2 + d^\epsilon$, so with probability at least $1 - k$ we have that $n \cdot |\langle \mathbf{x}, \mathbf{y} \rangle| = \frac{o(d)}{\|\mu\|^2 + d^\epsilon} \cdot |\langle \mathbf{x}, \mathbf{y} \rangle| \leq o(d)$. We also know from Lemma D.3 that with probability at least $1 - k$ we have that $\|\mathbf{x}\|^2 = \Omega(d)$. Setting $\tau = k \cdot n^2 = o_d(1)$ completes the proof. $\qquad \square$

**Mixture of $k$ Gaussians** We prove the case where we have 2 Gaussians, but the proof is similar for any number of Gaussians.

**Lemma D.6.** *Let $\mathcal{N} = \pi\mathcal{N}(\boldsymbol{\mu}^{(1)}, I) + (1 - \pi)\mathcal{N}(\boldsymbol{\mu}^{(2)}, I)$ where $0 \leq \pi \leq 1$ be a mixture of normal distributions on $\mathbb{R}^d$. Assume the following:*

- $\|\boldsymbol{\mu}^{(1)}\|^2 = o(d)$, $\|\boldsymbol{\mu}^{(2)}\|^2 = o(d)$

- $n = \frac{o(d)}{\max(\|\mu^{(1)}\|^2, \|\mu^{(2)}\|^2) + d^\epsilon}$ *for $\frac{1}{2} < \epsilon < 1$.*

- $k$ *defined as in Lemma D.5*

- $\tau = k \cdot n^2$

*then with probability at least $1 - \frac{\tau}{n^2}$ we have $n \cdot |\langle \mathbf{x}, \mathbf{y} \rangle| = o(d)$ and $\|\mathbf{x}\|^2 = O(d)$*

*Proof.* Let $\mathbf{x}, \mathbf{y} \sim \pi \mathcal{N}(\boldsymbol{\mu}^{(1)}, I) + (1 - \pi)\mathcal{N}(\boldsymbol{\mu}^{(2)}, I)$ where $0 \le \pi \le 1$. Let us compute $E\left[\|\mathbf{x}\|^2\right]$. We can think of $\mathbf{x}$ as

$$\mathbf{x} = \begin{cases} \mathbf{x}_1, & \text{with probability } \pi \\ \mathbf{x}_2, & \text{with probability } 1 - \pi \end{cases}$$

where $\mathbf{x}_1 \sim \mathcal{N}(\boldsymbol{\mu}^{(1)}, I)$ and $\mathbf{x}_2 \sim \mathcal{N}(\boldsymbol{\mu}^{(2)}, I)$. From the law of total expectation we get

$$E[\|\mathbf{x}\|^2] = \pi E[\|\mathbf{x}_1\|^2] + (1 - \pi)E[\|\mathbf{x}_2\|^2]$$

and from D.5 we get

$$E[\|\mathbf{x}\|^2] = \pi \cdot \left( \|\boldsymbol{\mu}^{(1)}\|^2 + \operatorname{tr}(I) \right) + (1 - \pi) \cdot \left( \|\boldsymbol{\mu}^{(2)}\|^2 + \operatorname{tr}(I) \right) = O(d)$$

Denote $A = \{\mathbf{x} : \left| \|\mathbf{x}\|^2 - E[\|\mathbf{x}\|]^2 \right| > d^\epsilon\}$ where $\frac{1}{2} < \epsilon < 1$.
From the law of total probability we get:

$$p(A) = p(A|\mathbf{x} = \mathbf{x}_1) \cdot \pi + p(A|\mathbf{x} = \mathbf{x}_2) \cdot (1 - \pi)$$

$$= 1 - \max\left( 2\exp\left( -\frac{c_1}{4c_2^2} \cdot d^{2\epsilon - 1} \right), 2\exp\left( -\frac{c_1}{2c_2^2} \cdot d^\epsilon \right) \right) - 2\exp\left( -\frac{c_1}{16c_2^2} \frac{d^{2\epsilon}}{\|\mu\|^2} \right) = 1 - o_d(1)$$

and specifically, $\|\mathbf{x}\|^2 = O(d)$.

Now, let us show that $E[\mathbf{x}^\top \mathbf{y}] = o(d)$:

$$E[\mathbf{x}^\top \mathbf{y}] = E[\mathbf{x}^\top]E[\mathbf{y}] = \left( \pi\boldsymbol{\mu}^{(1)} + (1 - \pi)\boldsymbol{\mu}^{(2)} \right)^\top \left( \pi\boldsymbol{\mu}^{(1)} + (1 - \pi)\boldsymbol{\mu}^{(2)} \right)$$

$$= \pi^2 \|\boldsymbol{\mu}^{(1)}\|^2 + 2\pi(1 - \pi)\boldsymbol{\mu}^{(1)\top} \boldsymbol{\mu}^{(2)} + (1 - \pi)^2 \|\boldsymbol{\mu}^{(2)}\|^2$$

$$= \pi^2 o(d) + 2\pi(1 - \pi)o(d) + (1 - \pi)^2 o(d) = o(d)$$

We divide the proof into 4 cases.

**Case 1**: $\mathbf{x}, \mathbf{y} \sim \mathcal{N}(\boldsymbol{\mu}^{(1)}, I)$

In this case, both points came from the same normal distribution, which we have already proven.

**Case 2**: $\mathbf{x} \sim \mathcal{N}(\boldsymbol{\mu}^{(1)}, I)$ and $\mathbf{y} \sim \mathcal{N}(\boldsymbol{\mu}^{(2)}, I)$

For every $i$ we have that $x_i$ and $y_i$ are sub-Gaussians and $\|x_i\|_{\psi_2} \le c$, $\|y_i\|_{\psi_2} \le c$, so we can use the same logic as in Lemma D.3 do prove that $\mathbf{x}^\top \mathbf{y} = o(d)$ with the same probability.

**Case 3**: $\mathbf{x} \sim \mathcal{N}(\boldsymbol{\mu}^{(2)}, I)$ and $\mathbf{y} \sim \mathcal{N}(\boldsymbol{\mu}^{(1)}, I)$

Same as case 2.

**Case 4**: $\mathbf{x} \sim \mathcal{N}(\boldsymbol{\mu}^{(2)}, I)$ and $\mathbf{y} \sim \mathcal{N}(\boldsymbol{\mu}^{(2)}, I)$

Same as case 1

Similar to D.4, with probability at least $1 - k$ we have that

$$n \cdot \langle \mathbf{x}, \mathbf{y} \rangle \le n \cdot \max(\|\mu^{(1)}\|^2, \|\mu^{(2)}\|^2) + d^\epsilon = o(d) \Rightarrow n = \frac{o(d)}{\max\left\{ \|\boldsymbol{\mu}^{(1)}\|^2, \|\boldsymbol{\mu}^{(2)}\|^2 \right\} + d^\epsilon}$$

and also that $\|\mathbf{x}\|^2 = \Omega(d)$. Setting $\tau = k \cdot n^2 = o_d(1)$ completes the proof. $\qquad \square$

