# OpenReview forum: "Provable Privacy Attacks on Trained Shallow Neural Networks"
_ICLR.cc/2025/Conference — Submitted to ICLR 2025_

### Official Review · Reviewer_hVLs · 2024-10-27

**Soundness:** 2
**Presentation:** 2
**Contribution:** 2
**Rating:** 5
**Confidence:** 3

**Summary:**

This paper looks at the problem of data reconstruction, and especially at the setting empirically discussed in [1]. The goal of the paper is to theoretically prove that reconstruction attacks are provably possible in some settings. The paper looks at data reconstruction in the univariate case (samples are one-dimensional), and also discusses membership inference attacks in the multivariate setting. The paper concludes with some experimental evidence beyond the setting covered by the technical assumptions used in the section on the membership inference attacks.

[1] https://arxiv.org/abs/2206.07758

**Strengths:**

The topic of data reconstruction gained reasonable popularity recently, and it is true that its theoretical understanding is lacking behind its practice. A better theoretical control of the current recovery schemes can be beneficial for the fields of privacy and security and beyond. Thus, in my opinion, the research question addressed by this paper is well motivated.

The paper implicitely tackles deep questions on the implicit bias of gradient descent algorithms on classification tasks, as the ratio of training points lying on the margin as a function of the data dimensionality (see Fig. 1a), or the output of the model at test time (see Fig. 1bc), which can be associated to the generalization error of the model itself.

**Weaknesses:**

First, the paper remarks multiple times that this is the first step towards understanding theoretically reconstruction attacks. This is false [2, 3]. Notice that these works (and probably other works I am not aware of) consider a different setting than the one presented in this paper, but (at least) equally interesting. I invite the authors to rephrase their narrative (see, e.g., "first step" in line 32 and "all previous works are empirical" in line 61) and to tune down this side of the contribution. On a minor level, I would invite the authors to remark in introduction that the idea of using the implicit bias of gradient descent on ReLU networks to understand data recovery is not new (see second paragraph of the introduction), as it is the core motivating idea behind the main paper the authors refer to [1].

Another weakness, in my opinion, is the high number of strong and implicit assumptions. The paper implicitely uses Theorem 2.1 from previous work to define its Assumption 2.1. This theorem assumes the loss to be small at some time, which I believe is not necessarily true (maybe the data / number of parameters have to respect some other explicit assumption), and therefore hides in which exact settings Assumption 2.1 is reasonable to be made. Can the authors elaborate, maybe in light of the results obtained in previous work on Theorem 2.1?

Another quite strong assumption made in this work is the implicit upper-bound on the number of training samples deriving from Assumption 4.1 (which translates to $n \ll \sqrt d$ in the case of the sphere). I wonder if with this number of data, it might be impossible for the model to successfully learn the task, as the output of the model at test time will approach 0, justifying why the results in Section 4 work (see for example Corollary 4.3, where at test time the outputs are small with high probability). To verify this, it would be helpful if the authors could also plot the test accuracy of the trained model in Figure 1, which will most likely decrease with $d$ as well. If it turns out that the attack success is strongly related with the test error of the model, it would be an important point to raise during the discussion.

Regarding the univariate case, I would also say it is hard to tell the effective predictive power of the results for more realistic scenarios. This setting might be qualitatively very different from the high dimensional case, and it is also difficult to evaluate how strong is the assumption that the attacker has knowledge of the margin (see line 190). Besides, I found section 3.2 a bit hard to parse at first, and I see the same problem I raised above on the implicit assumptions: what does it mean that $\Phi(\theta, x)$ is a local minimum of Eq. (1)? In which space? Is this stronger than Assumption 2.1? As the authors do not seem to be lacking in space, I would recommend adding an additional Figure in this section, where the intuition behind their results is depicted as a plot (for example, a uniivariate plot of  $\Phi(\theta, x)$ and the corresponding intersections with the margin). On top of this, it would be very helpful a brief discussion on why the results are intuitively true (see, for example, Theorem 3.2, where the reasons behind thesis seem a bit obscure).


[2] https://arxiv.org/abs/2201.04845

[3] https://arxiv.org/abs/2212.03714

**Questions:**

I asked multiple questions in the weaknesses section. I am happy to reconsider my score during the discussion time.

---

> ### Author Response · Authors · 2024-11-20
> **Weaknesses**
>
> We thank the reviewer for their time and feedback.
>
> - We never claimed that our work is the first to provide provable privacy vulnerabilities in general, but rather it is the first to provide \emph{provable} privacy vulnerabilities in the specific setting which takes advantage of the \emph{implicit bias of homogeneous networks}. The work of [1] prompted much further work, however these are, to the best of our knowledge, all empirical. We will further clarify this to avoid confusion.
>
> - As we previously mentioned in our reply to reviwer pxGc, our goal is to investigate the privacy implications of the implicit bias of trained neural networks. While minimization of the training loss is in general not guaranteed, this is less interesting from a practical perspective which strives to study the above question. Rather, we would like to analyze the privacy implications when optimization was successful, i.e., when the trained network has good accuracy. Theorem 2.1 provides a commonly used characterization of a successfully trained neural network, whose implication constitutes our baseline assumption. Such an assumption (that we have reached small training loss and thus converged in direction to a network satisfying the KKT conditions) is commonly used in the literature (e.g.\ [2-6]).
>
>     Analyzing whether gradient flow converges to a small training loss is an interesting theoretical problem, but the literature on implicit bias separates between this question, and the analysis of the properties of the trained network assuming a small training loss is achieved. In light of this, we believe that Assumption 2.1 is not particularly limiting.
>
> - The reviewer raises a valid concern regarding the generalization performance under our assumption.
>     For this reason, we have exemplified a mixture of Gaussians (line 341) as a distribution which satisfies Assumption 4.1. This, as explained in line 346, provides an example where our proposed attack is successful even when the generalization error is small.
>
> - Regarding the univariate case, this setting was also studied in multiple previous works (e.g.\ [6-9]). Unlike the high dimensional setting, here the sample size required for generalization is easier to control. The assumption that we have converged to a local minimizer of the margin maximization problem rather than just a stationary point is indeed stronger. This is discussed in line 290 in the paper. Currently, the only examples that we are familiar with where we converge to a non-local minimum of Eq. (1) are when initializing the neural network from a certain set of Lebesgue measure zero. This seems intuitively analogous to the fact that gradient descent converges to local minima of the objective, unless when initialized from a set of measure zero [10]. However, to the best of our knowledge, there is no such analogous result for the constrained optimization problem in Eq. (1). Regarding the margin being known to the attacker, it is not straightforward to deduce its value. It is difficult to gauge whether knowledge of the margin is a strong assumption or not, without making any further assumptions on the problem. We do stress however, as also discussed in the paper in line 172, that in the worst case, without any knowledge on the margin, this is still merely just a single hyperparameter that the attacker must accommodate for (e.g., the attacker can run our algorithm with various different guesses for the true value of the margin).
>
>     We will consider further explaining the intuition behind this result by using figures as the reviewer proposed.

---

> > ### Author Response · Authors · 2024-11-20
> > **References**
> >
> > 1. Niv Haim, Gal Vardi, Gilad Yehudai, Ohad Shamir, and Michal Irani. Reconstructing training data from trained neural networks
> >
> > 2. Gal Vardi, Gilad Yehudai, and Ohad Shamir. Gradient methods provably converge to non-robust networks
> >
> > 3. Spencer Frei, Gal Vardi, Peter Bartlett, and Nathan Srebro. Benign overfitting in linear classifiers and leaky relu networks from kkt conditions for margin maximization
> >
> > 4. Spencer Frei, Gal Vardi, Peter L. Bartlett, and Nathan Srebro. The double-edged sword of implicit bias: Generalization vs. robustness in relu networks
> >
> > 5. Guy Kornowski, Gilad Yehudai, and Ohad Shamir. From tempered to benign overfitting in relu neural networks
> >
> > 6. Itay Safran, Gal Vardi, and Jason D Lee. On the effective number of linear regions in shallow univariate relu networks: Convergence guarantees and implicit bias
> >
> > 7. Francis Williams, Matthew Trager, Daniele Panozzo, Claudio Silva, Denis Zorin, and Joan Bruna. Gradient dynamics of shallow univariate relu
> >
> > 8. Nirmit Joshi, Gal Vardi, and Nathan Srebro. Noisy interpolation learning with shallow univariate relu networks
> >
> > 9. Justin Sahs, Ryan Pyle, Aneel Damaraju, Josue Ortega Caro, Onur Tavaslioglu, Andy Lu, Fabio Anselmi, and Ankit B Patel. Shallow univariate relu networks as splines: Initialization, loss surface, hessian, and gradient flow dynamics
> >
> > 10. Jason D Lee, Max Simchowitz, Michael I Jordan, and Benjamin Recht. Gradient descent only converges to minimizers

---

> > ### Comment · Reviewer_hVLs · 2024-11-21
> >
> > I thank the authors for their rebuttal, I will comment on their raised points below.
> >
> > The authors stated that they are _the first to provide provable privacy vulnerabilities in the specific setting which takes advantage of the implicit bias of homogeneous networks_. In the narrative of this paper, this remark on this specific setting is missing or left implicit (see "this phenomenon" or "this setting" in the two lines I mentioned above), and gives the impression of a stronger theoretical contribution that what I believe the paper is bringing.
> >
> > More importantly, I understand and agree that the authors are not directly interested in the conditions for which optimization is feasible. However, this paper is rich in implicit (and explicit) assumptions, sometimes called from other related work, and it is hard to track what is the exact setting this paper can successfully cover (this comment is also for the last two replies of the authors). In this regard, I thank the authors for pointing out the example on the mixture of Gaussians where learning should be feasible due the results in Frei et al. 2023a. However, I would kindly ask if the authors can point a more precise point in that paper that covers the setting they consider, and if this discussion can be included in the paper. Furthermore, the experiments shown at the end of the paper only regard data uniformly distributed on the sphere, for which my point above should hold, that makes me believe the plots are focusing on a regime where there is no effective learnability. This to me is not a necessarily negative aspect of the analysis, but I really believe it is important to deliver a complete message here...

---

> > > ### Author Response · Authors · 2024-11-22
> > >
> > > We thank the reviewer for their response.
> > >
> > > In the revised version, we will clarify what is the specific setting in which we provide the first theoretical analysis, so that it is stated explicitly.
> > >
> > > The precise point in Frei et al. 2023a is the discussion after Theorem 5.2 (in their arXiv version). The authors mention there two specific settings that satisfy their theorem requirements, and thus good generalization performance can be achieved (and more specifically, in Corollary 5.3 they further show that in these settings good generalization is achieved by the max-margin linear predictor). Note that these settings satisfy our condition from lines 341-344.
> > >
> > > In addition, we would like to point to another paper that shows generalization in a similar setting: See the discussion after Theorem 4  in [1] (in the arXiv version). Their result implies generalization (for a max-margin linear classifier) in a setting where the data distribution consists of a mixture of two Gaussians (in $\mathbb{R}^d$) with covariance $I_d$ and means $\pm \mu$ where $\| \mu \| = d^\beta$ for $\beta \in (0.25,0.5)$, and their sample size is $n = \tilde{\Omega}(1)$. This setting satisfies our condition from lines 341-344. Generalization results for a similar distribution (a setting which satisfies our assumption) but for different models were shown in [2] (Theorem 3.1 in their arXiv version) and [3] (Theorem 3.1, conference version). Finally, generalization under our conditions from lines 341-344 was also shown in [4] (Theorem 3.1 in their arXiv version) for a mixture of four Gaussians (an XOR distribution).
> > >
> > > In the revised version, we will add an appendix where these settings are defined explicitly, to keep the paper self contained.
> > >
> > > Indeed, in our experiments the data points are uniformly distributed on the sphere. Since our experiments consider the vulnerability to membership inference attacks and do not consider generalization, we preferred to focus on this simpler setting. In the revised version we will discuss this issue, and add in the appendix similar experiments in a setting which also exhibits good generalization performance.
> > >
> > > We thank the reviewer for their constructive comments that help us improve our paper. If the reviewer has any additional concerns we will be happy to address them.
> > >
> > > [1] N. S. Chatterji and P. M. Long. Finite-sample Analysis of Interpolating Linear Classifiers in the Overparameterized Regime.
> > >
> > > [2] S. Frei, N. S. Chatterji and P. L. Bartlett. Benign Overfitting without Linearity:
> > > Neural Network Classifiers Trained by Gradient Descent for Noisy Linear Data.
> > >
> > > [3] X. Xu and Y. Gu. Benign overfitting of non-smooth neural networks beyond lazy training.
> > >
> > > [4] Z. Xu, Y. Wang, S. Frei, G. Vardi, W. Hu. Benign Overfitting and Grokking in ReLU Networks for XOR Cluster Data.

---

> > > ### Author Response · Authors · 2024-11-28
> > > **Revised version**
> > >
> > > Following up on our previous message, we would like to update you that we have uploaded a revised version. For details about the modification, please see our official comment to the paper itself.

---

> > > > ### Comment · Reviewer_hVLs · 2024-12-02
> > > >
> > > > I thank the authors for their response and update, and I apologize for the late feedback. I would like to ask one more question, if the authors can find time before the end of the discussion period.
> > > >
> > > > I was reading the newly written appendix C, in particular with respect to the comparison with Frei et. al 2023. I am reading their Theorem 11 and Corollaries 12 and 13. In all these statements, there are a lot of moving parts, in particular a strict lower bound on $n$ to indeed make the statement of Theorem 11 to not be vacuous (i.e. the conditions to have benign overfitting).
> > > >
> > > > As far as I understood, in the revision there is no comment on such lower bound, and the reasons behind the effective learnability are still (in my opinion) not crystal clear. Can the authors elaborate on that? Notice that this comment matches with my previous comment (in the original review) on the other implicit assumptions contained in the paper.
> > > >
> > > > In case of a resubmission, I strongly encourage the authors in motivating and remarking the effectiveness of the setting, in the most self-consistent fashion possible. In this regard, the lack of experimental evaluation on the generalization capabilities of the models (both classification and regression) further increases the difficulty in interpreting the results, and especially their relevance for broader settings (in this, I agree with Reviewer uP3P).

---

> > > > > ### Author Response · Authors · 2024-12-02
> > > > >
> > > > > Thanks for your response.
> > > > > The results in Frei et. al 2023 (and also in the other references that we provided in Appendix C) require lower bounds on the sample size, but these lower bounds are smaller than the upper bound in our Item 3 in lines 353-355, and hence their settings satisfy our assumption.
> > > > >
> > > > > Essentially, these benign-overfitting results require both upper and lower bounds on the sample size. Their assumptions on the upper bounds are roughly similar to our assumption (and sometimes even stronger), and their lower bounds are not required in our case (because our result about privacy holds both in the regime where there is good generalization and in the regime where the sample size it too small to guarantee good generalization).
> > > > >
> > > > > We will clarify this in the next revision as the reviewer suggested.

---

### Official Review · Reviewer_uP3P · 2024-10-29

**Soundness:** 2
**Presentation:** 2
**Contribution:** 2
**Rating:** 5
**Confidence:** 4

**Summary:**

This paper studies the privacy attack on 2-layer Relu activated neural network. Under the assumption that the model converges to a point that satisfies the KKT condition, this paper shows 2 attacks: 1) for the univariate case, this paper is able to recover a part of the training samples from the weights; 2) when the number of training samples $n$ is much smaller than the dimension $d$ of the training data, the output of the neural network can serve as membership inference attack. Experiments show that even n is as large as d, the membership inference attack can still work empirically.

**Strengths:**

This paper studies an important problem of privacy in neural networks. Theoretical analysis are provided. Experiments are conducted to support the theoretical claims. Source codes are provided.

**Weaknesses:**

My biggest concern is that the novelty seems to be limited.
- It is well known [1] that overparameterized neural networks can memorize training input, and it is not surprising that the training input is encoded in the model weights in some way. Actually, empirical study has shown that it is possible to reconstruct training data from model weights [2] and even model updates [3]. From this perspective, the contribution of this work seems to be limited at providing a theoretical proof. However, the proposed attack only work for the univariate case, which means the neural network is essentially a piece-wise linear function. It is not clear how this work can inspire training data reconstruction for general functions.
- For the membership inference attack, in the setting where $n \ll d$, we can think that the model is able to memorize the label ($\pm 1
$) for all the $n$ samples that are seen during training, but the model does not have any idea for the remaining input , which is at least a $d-n$-dimensional subspace. So it is expected that the model output can serve as an indicator for membership. The condition on $n,d$ and the assumption of almost orthogonal training data seem to be quite limited. It is not clear if the proposed theory can help understand the membership inference attack in general case, e.g. [4].

[1] Allen-Zhu, Zeyuan, Yuanzhi Li, and Zhao Song. "A convergence theory for deep learning via over-parameterization." International conference on machine learning. PMLR, 2019.

[2] Haim, Niv, et al. "Reconstructing training data from trained neural networks." Advances in Neural Information Processing Systems 35 (2022): 22911-22924.

[3] Wang, Zihan, Jason Lee, and Qi Lei. "Reconstructing training data from model gradient, provably." International Conference on Artificial Intelligence and Statistics. PMLR, 2023.

[4] Sablayrolles, Alexandre, et al. "White-box vs black-box: Bayes optimal strategies for membership inference." International Conference on Machine Learning. PMLR, 2019.

**Questions:**

- The curve in Figure 1 (a) is not very smooth. If you run multiple experiments and report the average, will it be better?
- Loss is a popular metric for membership inference attack. In general trained samples shall have smaller losses than test samples. Can you explain a bit more on the relationship between loss and margin?

---

> ### Author Response · Authors · 2024-11-20
> **Concerns regarding novelty**
>
> We thank the reviewer for their time and feedback.
>
> - As the reviewer observed, our main contribution in this paper is to provide a rigorous theoretical analysis in a setting similar to that which was empirically used in [1]. The reference given in [2], which the reviewer mentioned, deals with a theoretical setting, but one which is different than ours (namely, they considered a setting where the gradients are known to the attacker). In particular, to the best of our knowledge, our work is the first to show provable privacy attacks in the setting considered in [1]. Although it is well known that over-parameterized neural networks can memorize training input, it is not immediately clear how to use this in order to extract the samples from a trained network and when it is at all possible.
>
>     We believe that rigorous results might provide useful insights on this phenomenon.
>     We agree that the univariate setting is very restricted,
>     however, many previous works have considered a similar setting [1-4]. Lastly, since this is the first provable privacy vulnerability result in this setting, we believe that it could prove to be a stepping stone in obtaining more such results.
>
> - We understand the reviewer's intuition for the proof of Theorem 4.2, however we also point out that rigorously proving the claim is highly non-trivial, as evident in our proofs. Our purpose in this paper, as in many theoretical papers, is to provide proofs rather than just intuitions, to elucidate the root cause for these vulnerabilities, which will motivate further research and lay the foundation for provable \emph{defenses}.
>
>     We would also like to point our that a similar assumption to Assumption 4.1 was made in many recent papers, e.g., [6-9].
>     so from a theoretical perspective this assumption is quite standard.

---

> > ### Author Response · Authors · 2024-11-20
> > **Questions**
> >
> > - As suggested, we ran more experiments which indeed smooth out the plot. We will update this in the revised version of our paper.
> >
> > - In our setting, intuitively, the more the neural network is trained, the smaller the loss will be on each individual example, and the more the margin of the network will increase. As revealed by Theorem 4.2, this will increase the gap between the prediction value of points in the training set and points in the test set, which is in keeping with the reviewer's comment regarding the loss being a popular metric for MIAs.

---

> > > ### Author Response · Authors · 2024-11-20
> > > **References**
> > >
> > > 1. Niv Haim, Gal Vardi, Gilad Yehudai, Ohad Shamir, and Michal Irani. Reconstructing training data from trained neural networks
> > >
> > > 2. Zihan Wang, Jason Lee, and Qi Lei. Reconstructing training data from model gradient, provably
> > >
> > > 3. Itay Safran, Gal Vardi, and Jason D Lee. On the effective number of linear regions in shallow univariate relu networks: Convergence guarantees and implicit bias
> > >
> > > 4. Francis Williams, Matthew Trager, Daniele Panozzo, Claudio Silva, Denis Zorin, and Joan Bruna. Gradient dynamics of shallow univariate relu networks.
> > >
> > > 5. Justin Sahs, Ryan Pyle, Aneel Damaraju, Josue Ortega Caro, Onur Tavaslioglu, Andy Lu, Fabio Anselmi, and Ankit B Patel. Shallow univariate relu networks as splines: Initialization, loss surface, hessian, and gradient flow dynamics
> > >
> > > 6. Gal Vardi, Gilad Yehudai, and Ohad Shamir. Gradient methods provably converge to non-robust networks
> > >
> > > 7. Spencer Frei and Gal Vardi and Peter L. Bartlett and Nathan Srebro and Wei Hu. Implicit Bias in Leaky ReLU Networks Trained on High-Dimensional Data
> > >
> > > 8. Guy Kornowski, Gilad Yehudai, and Ohad Shamir. From tempered to benign overfitting in relu neural networks
> > >
> > > 9. Spencer Frei, Gal Vardi, Peter Bartlett, and Nathan Srebro. Benign overfitting in linear classifiers and leaky relu networks from kkt conditions for margin maximization

---

> > ### Comment · Reviewer_uP3P · 2024-11-25
> >
> > I thank the authors for the rebuttal. Some of my questions are resolved, and I raise my evaluation accordingly. Though I still have concerns on the assumptions/settings in the paper. For the univariate setting, it remains unclear how the results in this restricted setting might generalize to broader and more practical scenarios. I understand that this setting has been studied in other works, but I am still not convinced how the proposed proof would help us better understand deeper networks/more features. For the high dimensional case, in particular Assumption 4.1, while I understand that similar assumptions have been used in prior work, I have significant reservations about its implications. As I mentioned this means only a tiny fraction of the input space is shown to the neural network, which makes membership inference attack appears easy to implement. Also this is very different from practical dataset settings.

---

### Official Review · Reviewer_QvBF · 2024-11-01

**Soundness:** 3
**Presentation:** 2
**Contribution:** 4
**Rating:** 8
**Confidence:** 3

**Summary:**

This paper describes how 2-layer neural networks have provably effective privacy attacks. The reconstruction attack they instantiate assumes complete optimization to the implicit bias solution and then analyzes the implication of the KKT conditions in cases to deduce high probability sets of training datapoints. Notably this algorithm does not require any further assumptions on the dataset, and hence presents a strong attack. The membership inference attacks impose certain restrictions on the data generating distribution, which I will summarize as “being sufficiently spread out in direction and magnitude” (i.e., orthogonal with high probability, and high magnitude relative to dimension), and is the case for common distributions. All together, these are (to the best of my knowledge) the first results on general privacy attacks against deep neural networks, and I believe also presents questions for future empirical and theoretical study.

**Strengths:**

1) As far as I am aware, the first general results of effective privacy attacks on neural networks, as opposed to empirical results known for specific datasets and models.
2) Surfacing assumptions required, which future work can weaken; the paper already discusses how the dimensionality assumptions may be weakened (given empirical evidence).
3) I believe the techniques in the proofs for the attacks to be novel to the private ML community

**Weaknesses:**

1) I think the presentation could make it more explicit what the contributions of the theory is to the broader privacy community. I attribute this mainly to the use of vague language regarding the assumptions in the main text, such as in line 45 which states “certain settings with varying assumptions”. One could instead highlight specific assumptions (e.g., dimensionality for membership inference or the data-independence of the reconstruction attacks) and contrast them with what is already known empirically.

2) On presentation, I had some questions for the proofs as presented, which I believe are resolvable but I list them below in the Questions section.

**Questions:**

1) In the proof of Theorem 4.2, in lines 117-118 it is not clear to me how this follows without a high probability statement. The theorem states that with probability $1- \tau$ you have the training points are on the margin, and I believe some high probability argument like Lemma B.3 is implicit in the statement (and would be great to state more explicitly in the proof for completeness).


2) Similarly, at the end of the proof of Theorem 4.2, I believe you mean the final equality occurs with probability $1- 2\tau$ not $1 - \tau$.


3) In the proof of Theorem 3.2 and the existence of $x_1$, it is not clear to me how it not existing would imply the break points were the same with the stated logic. Could we not have $w_{i-1}$ be positive and hence it is determined by the margin points after the breakpoint, but $w_{i}$ can be negative and hence determined by the margin point below its breakpoint (which by assumption would be $0$). In this way they are still different given that $x_2$ exists, which is necessary as there is at least one margin point? What seems to me to be the correct proof is now noting that the breakpoint for $i+1$ would be the same as either $i$ or $i-1$, leading to a contradiction and hence needing $x_1$ to exist.


4) In Line 143 I believe you mean to say “for the sake succinctness” not “completeness” as you are summarizing the statement?

---

> ### Author Response · Authors · 2024-11-20
> **Weaknesses And Questions**
>
> Thank you for the helpful comment, we will further clarify these issues in the revised version of the paper.
>
> Question 1
>
> The reviewer is correct. The point is on the margin when $m - (\delta + 1)(n-1) \frac{m}{\Delta + m - 2\delta(n-1)} > 0$. This happens when $\Delta > (2\frac{\delta}{\delta + 1} + 1)(\delta + 1)(n-1)$, which holds with high probability. Following the reviewer's comment, we also fixed a minor bug which was due to a typo in the proof of Lemma~B.2, where we replaced $m$ with $1$.
>
> Question 2
>
> Thanks for catching this, you are correct and we will fix this.
>
> Question 3
>
> The reviewer is correct again, and the proposed solution indeed works. We will change the proof accordingly.
>
> Question 4
>
> By ``sake of completeness," we meant that we present the theorem here to keep the paper self-contained.

---

> > ### Comment · Reviewer_QvBF · 2024-11-22
> >
> > I thank the reviewers for their clarifications. At this moment I have no other technical concerns, but wish to point out presentation of assumptions and settings is still a concern (and from what I see a common one amongst the other reviewers).
> >
> > I suggest that the authors consider implementing some changes as revisions to the manuscript before the discussion phase is over, as to get feedback from reviewers on if the changes are satisfactory. For example, modifications to the specific phrasing I raised in the introduction (or elsewhere as raised by other reviewers).

---

### Official Review · Reviewer_pxGc · 2024-11-03

**Soundness:** 2
**Presentation:** 1
**Contribution:** 3
**Rating:** 6
**Confidence:** 3

**Summary:**

The paper tries to show provable guarantees for data reconstruction and membership inference for nonconvex activation functions in shallow Neural networks.

**Strengths:**

The overall question is interesting and would certainly be of interest to the community at large. The main contribution of this paper is to show there is _some_ theoretical basis for the literature on data reconstruction and membership inference attacks. However, the results in this paper are somewhat lacklustre for reasons outlined in the next two sections.

**Weaknesses:**

The paper is lacking in the areas of exposition and technical details. Concepts which are central to the paper such as Gradient flow, Data Reconstruction, and Membership Inference are never formally defined, which makes the paper inaccessible to all but a very niche audience.

I have listed a set of points in the next section for the authors to clarify. In the interest of transparency, I am clearly stating my position here:
  - The paper needs more work especially in exposition (see above). At this time it reads like a first draft. For example, the assumptions and the proofs are tightly coupled with each other, without a clear basis for either. Simply based on this issue, I am recommending rejection for the paper.
  - I am open to changing my score in case I have misunderstood something regarding the assumptions. However, I again stress that such clarifications should have been in the paper to begin with, and it will probably not improve my score by a lot.

*Remark*: Judging by the previous works of Haim et al. (2023) etc. it seems to be that it is indeed possible to reconstruct data or perform MI in the settings considered. I believe that the paper stands to benefit a lot if the focus of the theoretical setting would have been to investigate the precise nature of the assumptions that allowed these attacks to take place.

**Questions:**

1. Theorem 2.1 specifically mentions that the gradient flow in a homogenous ReLU NN _eventually converges_  (i.e., at $t\to\infty$) to a stationary point.
  - Line 146: What does gradient flow over a binary classification set formally mean?
  - Line 154: I invite the authors to explain why it is natural to assume that the existence of a point of convergence automatically implies that the function has converged. It might be a fine assumption for experimentally focused papers, but assuming convergence in a theoretical setting needs to be well motivated.
  - Does the above issue not mean that Assumption 2.1 is an extremely strong condition for a 2-layer NN?

2. I think that the proof of Theorem 3.1 is incorrect (or at the very least missing certain assertions).
  - it is stated that the real-valued function $\phi(\theta; x_1)$ cannot be the zero function because for all $i\in[n]$, $y_i\phi(\theta;x_i)\geq m$, where $m:=\min_i|{\phi(\theta;x_i)}|$, and therefore can well be $0$. The authors are therefore assuming another implicit condition, i.e., the margin should always be positive. This makes Assumption 2.1 even more untenable for 2-layer NNs in the context of Theorem 2.1.
  - Suppose $\phi(\theta; x_1)>0$. Since $\phi$ is a real-valued function, I am not understanding why $y_1\Phi(\theta;x_1)$ cannot lie in between 0 and 1. Hence the rest of the implications in the proof do not follow.
   - The crux of Assumption 2.1 should be clearly stated in the abstract of the paper and the introduction of the paper in some form.
As it stands, I am not convinced that even in the univariate case, the data can be recovered.

3. In Line 309, the authors state: "In very high dimensional settings, under many commonly used data distributions, we have that the dataset is almost orthogonal with high probability." This forms the basis for Assumption 4.1, which is critical to the rest of the proofs. This is the first time the authors have mentioned the ability to perform membership inference or data reconstruction w.r.t. well-behaved distributions. This is highly unusual and subtracts from the blackbox nature of membership inference or data reconstruction attacks.
   - For Assumption 4.1 to be considered _valid_, I would prefer if the authors included a discussion addressing the above concern.
   - The crux of Assumption 4.1 should be clearly stated in the abstract of the paper and the introduction of the paper in some form.

4. Most of the main theorems are missing a high level overview from the main paper. The authors should include a proof sketch of Theorems 3.2 and 3.3.

5. Line 474: I invite the authors to clarify the following statement - "in this section, we conducted a few experiments focusing on the membership inference problem, and observed that while our theoretical results’ assumptions do not necessarily hold, their implications are nevertheless still valid." Does this not mean that weaker assumptions would have sufficed?


## Typos and Grammatical mistakes
Line 231: "be two adjacent intervals which none of them is constant on the margin" - the authors should rewrite this sentence.

---

> ### Author Response · Authors · 2024-11-20
> **Weaknesses**
>
> We thank the reviewer for their time and feedback.
>
> Regarding the lack of formal definitions for several concepts:
> The notions "data reconstruction" and "membership inference" can have different formal definitions, depending on the considered setting. In our theorems, we formally specify what we mean by data reconstruction and membership inference.
> Gradient flow is defined informally in line 146 as "a continuous time analog of gradient descent". We did not provide a formal definition since we thought it's already quite standard in the literature these days, but, as the reviewer suggested, we will add a formal definition (see our response to question 1 below).
>
> We do not understand what the reviewer means by "the assumptions and the proofs are tightly coupled with each other". Is there an assumption that the reviewer views as unreasonable or unclear (please see our responses to your specific questions below)?
>
> The reviewer wrote: "I believe that the paper stands to benefit a lot if the focus of the theoretical setting would have been to investigate the precise nature of the assumptions that allowed these attacks to take place". We agree that rigorous proof for reconstruction in the setting of Haim et al. would be fantastic. However, it is usually the case in theoretical works that the considered setting is much more limited than what can be demonstrated in practice. Expecting a rigorous proof that follows the setting of Haim et al. seems unrealistic.

---

> ### Author Response · Authors · 2024-11-20
> **Questions**
>
> Question 1
>
>     1. Line 146: This means that we optimize the empirical training loss (as defined in line 135) using gradient flow, which corresponds to gradient descent with an infinitesimal step size:
>
>     Let $\btheta(t)$ be the trajectory of gradient flow. Starting from an initial point $\btheta(0)$, the dynamics of $\btheta(t)$ is given by the differential equation
>     $\frac{d \btheta(t)}{dt} = -\nabla L(\btheta(t))$.
>     (Or more formally, $\frac{d \btheta(t)}{dt} \in -\partial L(\btheta(t))$, where $\partial$ denotes the \emph{Clarke subdifferential}, which is a generalization of the derivative for non-differentiable functions.)
>     We will further clarify this in the paper.
>
>    2. Line 154: Our goal is to investigate the privacy implications of the implicit bias of trained neural networks. While minimization of the training loss is in general not guaranteed, this is less interesting from a practical perspective which strives to study the above question. Rather, we would like to analyze the privacy implications when optimization was successful, i.e., when the trained network has good accuracy. Theorem 2.1 provides a commonly used characterization of a successfully trained neural network, whose implication constitutes our baseline assumption. Such an assumption (that we have reached small training loss and thus converged in direction to a network satisfying the KKT conditions) is commonly used in the literature (e.g.\ [1-5]).
>
>     Analyzing whether gradient flow converges to a small training loss is an interesting theoretical problem, but the literature on implicit bias separates between this question, and the analysis of the properties of the trained network assuming a small training loss is achieved. In light of this, we believe that Assumption 2.1 is not particularly limiting.
>
> Question 2
>
>     1. Following our answer to Question 1 above, we believe that the assumption that $m>0$ is well justified. The reviewer is correct to point out that this should be stated explicitly, and we will therefore correct this.
>     2. Throughout this section, we assumed w.l.o.g. that the value of the margin is 1, which implies that $y_1 \Phi(\btheta, \bx_1) = 1$.
>
>     3. Our abstract explicitly states that ``We prove that theoretical results on the implicit bias of 2-layer neural networks...", which refers to Assumption 2.1. We are not sure if the reviewer is proposing to explicitly state that we converge to a KKT point, but in any case this is a definition we would not expect most readers to be familiar with, and therefore we decided to not include it in the abstract.
>
> Question 3
>
>     We are not sure that we understand what the reviewer means by ``blackbox nature of membership inference". In the membership-inference literature, it is common to assume that the attacker has knowledge on the input distribution. For example, in the survey [6] on membership inference attacks (MIA) which is cited in our paper, the author mentions in section 3.2 some common assumptions about the knowledge that the attacker has on the distribution. In particular, it is mentioned that ``In most settings of MIAs, the distribution of training data is assumed to be available to an attacker of MIA". As in the previous bullet point in Question 2, we are not sure what the reviewer means by the crux of the assumption, but we
>     mentioned both in the abstract and the introduction that this result considers a "high-dimensional setting".
>
> Question 4
>
>    We thank the author for this comment. We will provide a sketch as proposed (within the allowed space).
>
> Question 5
>
>     We believe that it might be possible to slightly relax our assumptions on the relation between $d$ and $n$ and derive the same results, but this might require a significant additional technical overhead.

---

> > ### Author Response · Authors · 2024-11-20
> > **References**
> >
> > 1. Gal Vardi, Gilad Yehudai, and Ohad Shamir. Gradient methods provably converge to non-robust networks
> >
> > 2. Spencer Frei, Gal Vardi, Peter Bartlett, and Nathan Srebro. Benign overfitting in linear classifiers and leaky relu networks from kkt conditions for margin maximization
> >
> > 3. Spencer Frei, Gal Vardi, Peter L. Bartlett, and Nathan Srebro. The double-edged sword of implicit bias: Generalization vs. robustness in relu networks
> >
> > 4. Itay Safran, Gal Vardi, and Jason D Lee. On the effective number of linear regions in shallow univariate relu networks: Convergence guarantees and implicit bias
> >
> > 5. Guy Kornowski, Gilad Yehudai, and Ohad Shamir. From tempered to benign overfitting in relu neural networks
> >
> > 6. Hongsheng Hu, Zoran Salcic, Lichao Sun, Gillian Dobbie, Philip S Yu, and Xuyun Zhang. Membership inference attacks on machine learning: A survey.

---

> > ### Comment · Reviewer_pxGc · 2024-11-21
> > **Additional comments to the authors**
> >
> > #Question 1
> > -----
> >
> > >  Our goal is to investigate the privacy implications of the implicit bias of trained neural networks. While minimization of the training loss is in general not guaranteed, this is less interesting from a practical perspective which strives to study the above question. Rather, we would like to analyze the privacy implications when optimization was successful, i.e., when the trained network has good accuracy.
> >
> > I thank the authors for this clarification. I had indeed missed this subtlety earlier, and the paper suddenly makes a whole lot more sense to me.  It seems that the goal of the paper is to provide a more theoretical basis for empirical observations, i.e., _what provable privacy attacks can be shown on *trained* NN_. I am including this comment here, since I believe it might be of help for others who might also miss the distinction.
> >
> > #Question 2
> > ----
> >
> > > 1.... we believe that the assumption that $m>0$ is well justified. The reviewer is correct to point out that this should be stated explicitly, and we will therefore correct this.
> >
> > I agree with this point.
> >
> > > 2. Throughout this section, we assumed w.l.o.g. that the value of the margin is 1, which implies that $y_1 \Phi(\theta, x_1) = 1$.
> >
> > Assuming without loss of generality that the margin induced on training samples is 1 seemed quite a strong assumption for me earlier, especially considering the fact that there are areas of research devoted to amplifying small margins (as small as $\frac{1}{\mathrm{poly}{m}$) - e.g., boosting algorithms. I agree that this assumption also follows from the fact that these are properties of *trained* NNs.
> >
> > I would recommend that both of these assumptions should be more prominently highlighted in the manuscript, maybe as Assumptions 2.2 and Assumptions 2.3.
> >
> > That being said, the proof is indeed correct conditioned on the above two assumptions.
> >
> > Additional Comments
> > ---------
> > 1. Maybe consider changing "Assumptions" to "Settings". From what I understand, there exist trained NN under these heuristic assumptions, that are not theoretically robust to a large umbrella of privacy attacks as evidenced by the results in this paper. In this case, the current paper does not bear the load of justifying the so-called assumptions.
> > 2. I am still hesitant about the general presentation of the paper, and I believe that the results can be presented better. This is a personal opinion, and I do not expect the authors to be able to address this issue. However, I am changing my score to reflect a positive opinion about the paper.

---

### Author Response · Authors · 2024-11-28
**Revised version**

We thank we reviewers for their comments and time.
We have uploaded a revised version with the following modifications:
- We emphasized in the introduction and background sections, the specific settings and assumptions we use and explained better why they are reasonable to make.
- We added short proof sketches of Theorem 3.2 and Theorem 3.3.
- We added a discussion in the appendix regarding the learnable cases that satisfy Assumption 4.1 (see Appendix C and a reference in line 358).
While we have not added to the current version more experiments, we are running identical experiments as in Section 5, but for a mixture of two Gaussians. The preliminary results look promising. We will add it to the next revision.

If the reviewer has any additional concerns, we will be happy to address them

---

### Meta-Review · Area_Chair_f5pA · 2024-12-22

**Metareview:**

This paper studies the privacy attack on 2-layer Relu activated neural network. Under the assumption that the model converges to a point that satisfies the KKT condition, this paper shows 2 attacks: 1) for the univariate case, this paper is able to recover a part of the training samples from the weights; 2) when the number of training samples is much smaller than the dimension of the training data, the output of the neural network can serve as membership inference attack. Experiments show that even n is as large as d, the membership inference attack can still work empirically.

This paper is on the borderline. I agree with the two of the reviewers regarding abstaining from full acceptance, due to the difficulty in the generalization capabilities of the models and their relevance for broader settings.

**Additional Comments On Reviewer Discussion:**

In case of a resubmission, I strongly encourage the authors in motivating and remarking the effectiveness of the setting, in the most self-consistent fashion possible. In this regard, the lack of experimental evaluation on the generalization capabilities of the models (both classification and regression) further increases the difficulty in interpreting the results, and especially their relevance for broader settings (in this, I agree with Reviewer uP3P).

---

### Decision · Program_Chairs · 2025-01-22

Reject